# TRIDENT: Cross-Domain Trajectory Spatio-Temporal Representation via Distance-Preserving Triplet Learning

**Guan-Yi Jhang[1], Jeng-Chung Lien[2], Hui-Ching Yu[1], Hsu-Chao Lai[3], Jiun-Long Huang[1]**
[1]National Yang Ming Chiao Tung University    [2]The University of Queensland    [3]Chang Gung University
{jhangguanyi.c, lillian.hc.yu.cs11, jlhuang}@nycu.edu.tw,
jengchung.lien@gmail.com, hsuchaolai@cgu.edu.tw

## Abstract

We present the TRIplet-based Distance-preserving Embedding Network for Trajectories (TRIDENT), a spatio-temporal representation framework for compressing and retrieving trajectories across scales, from badminton courts to large-scale urban environments. Existing methods often assume smooth, continuous motion, but real trajectories exhibit event-driven annotation, abrupt direction changes, GPS errors, irregular sampling, and domain shifts, exposing the inefficiency, limited generalization, and inability to robustly integrate temporal order with spatial sequence structure of prior models. TRIDENT addresses these challenges by combining Graph Convolutional Network (GCN) spatial embeddings with temporal features in a Dual-Attention Encoder (DAEncoder), along with a Nonlinear Tanh-Projection Attention Pooling (NTAP) module that preserves local order and robustness under noise. For metric learning, we introduce a Distance-preserving Multi-kernel Triplet Loss (DMT) to preserve pairwise spatio-temporal distances in the native feature space and their rank order within the embedding, thereby reducing geometry distortion and improving cross-domain generalization. Experiments on urban mobility and badminton datasets show that TRIDENT outperforms strong baselines in retrieval accuracy, efficiency, and cross-domain generalization. Furthermore, the learned embeddings capture spatio-temporal sequence patterns, facilitating tactical analysis of badminton rallies via silhouette-guided spectral clustering that provides more actionable insights than direct trajectory classification.

## 1 Introduction

With the development of general imaging facilities and location infrastructure, large volumes of trajectory data are collected and used to capture human or vehicle movements (Hsu et al., 2019; Zheng, 2015), which results in various research emerging in trajectory analysis (Chang et al., 2023a; Si et al., 2024). Trajectory clustering is a core task in trajectory analysis. Its aim is to discover similar trajectories, which facilitate applications in hotspot detection (Fang et al., 2020), movement pattern mining (Chen et al., 2019), and abnormal activity prediction (Fernando et al., 2017). Despite this progress, challenges remain. Handcrafted distances and raw-data–based encodings at the point, segment, or polyline level struggle to generalize across datasets with different lengths, sampling rates, and motion patterns, and they trade off local spatial detail against global trajectory shape (e.g., EDR vs. Hausdorff). In contrast, deep learning approaches learn task-specific representations directly from data, which reduces dependence on predefined metrics and can improve generalization across dataset types. Furthermore, existing similarity measures either focus solely on local spatial dependencies (point-based) or emphasize only global relationships between trajectories (shape-based), which makes it challenging to capture both levels simultaneously. Finally, trajectory data often suffer from measurement and sampling issues that distort the paths and their derived features, which in turn degrades similarity estimation and clustering quality. When trajectories come from GPS sampled coordinates (Zheng, 2015), urban canyons weaken positioning signals and introduce large errors. If the sampling rate is low or irregular, the few available points cannot faithfully represent

the true movement (Li et al., 2018). When trajectories come from marker based video annotation in Wang et al. (2023b), differences in marker placement such as left or right foot and forefoot or heel lead to inter annotator variability and additional noise (Wang et al., 2023a).

In time sensitive applications such as traffic planning (Tran et al., 2020) and surveillance (Yu et al., 2019), trajectory analysis often involves continuous GPS traces (see Section 2). In space sensitive applications such as movement pattern mining (Chang et al., 2023a; Lucey et al., 2013), the trajectories of badminton players (Wang et al., 2023a) are discrete traces (see Section 2). We discuss these two settings together because they represent complementary challenges. Most prior work focuses on smooth driving data (Si et al., 2024; Fang et al., 2022; Chang et al., 2023b; Zhang et al., 2020; Yao et al., 2019), while sports trajectories remain under explored. To provide a foundational understanding, we present three figures that highlight key movement differences. Fig. 1a and 1b depict taxi trajectories collected in Beijing and Rome, each vehicle is equipped with a GPS device that samples and transmits location data (latitude and longitude) at 7-seconds intervals, yielding low-curvature trajectories. In contrast, Fig. 1c illustrates badminton players' movements at hit moments. Rapid, nonlinear motion patterns and high-frequency shifts pose challenges to modeling. The state-of-the-art (SOTA) trajectory clustering models fail to capture the underlying patterns in the Badminton dataset, revealing a gap that has limited progress on this topic despite the existence of such movement patterns (Appendix A.1). Currently, we encounter several challenges:

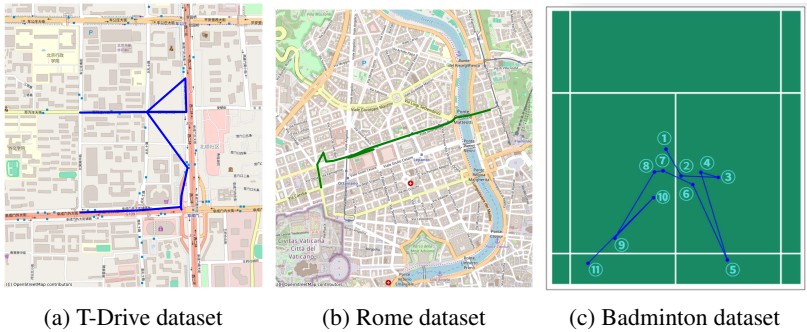

(a) T-Drive dataset       (b) Rome dataset       (c) Badminton dataset

Figure 1: Trajectory comparison among different datasets.

**(1) Scalable Unified Trajectory Representation across Sampling Regimes and Movement Styles:** Self-supervised modeling of badminton trajectories is challenging. Many trajectory models target densely sampled and smooth GPS traces, whereas research on badminton trajectories remains scarce due to their discrete characteristics. We use a single unified architecture that can be trained on different data types. This avoids redesigning distances and model components when data collection shifts from continuous to event-driven annotation. It also keeps nearest neighbor queries consistent within each dataset when sampling density or annotation style varies. Many emphasize only one aspect or raw coordinates representation, which limits tactical insight and reduces robustness.

**(2) Complexity of Trajectory Clustering**: Badminton trajectories combine discrete shot locations and continuous player coordinates with sequence dependencies. This combination increases the complexity of trajectory clustering, as traditional methods struggle to integrate spatial and temporal sequence attributes (Hamdi et al., 2022). Taxi trajectories suffer from GPS errors and interruptions, which induce drift and jumps that persist after map matching. Variable and missing sampling yields trajectories that are sparse or overly dense, hindering alignment. Device heterogeneity and cross-city variations in network density, traffic restrictions, and signal control lead to distinct distributions.

To address these challenges, we integrate multiple dataset into the TRIplet-based Distance-preserving Embedding Network for Trajectories (TRIDENT) to evaluate its effectiveness in distinct trajectory analysis. We use a Graph Convolutional Network(GCN) to capture spatial and Date2Vec(D2V) to capture temporal features. Using Dual-Attention Encoder (DAEncoder), we sequentially fuse temporal and spatial embeddings while preserving their cross-domain interactions and frequency-domain dependencies. In addition, we introduce the Nonlinear Tanh-Projection Attention Pooling (NTAP). By capturing higher-order nonlinear interactions and dynamically reweighting subsequences through a learnable context vector, NTAP resolves the discrete event-

driven structure commonly found in badminton trajectories, enabling consistent representation. We further propose Distance-preserving Multi-kernel Triplet Loss (DMT), a geometry-aware loss that prioritizes trajectories of varying lengths and uses a multi-kernel mechanism to accommodate diverse characteristics and scales across discrete and continuous feature spaces. The geometry is induced by the distances among the anchor, positive, and negative samples, directly tying the loss to the triplet objective, which we call TRIDENT. Our contributions are summarized as follows:

**(1) Generalist Trajectory Learning**: We propose a data driven approach for unified trajectory modeling and self-supervised clustering. To the best of our knowledge this is the first unified representation that can learn either continuous movement trajectories or discrete action sequence trajectories, and it achieves superior performance. The method reduces the bias toward continuous trajectory models, exemplified by smooth taxi trajectories, and works well on Badminton data, closing a key literature gap. Validation spans selected heterogeneous datasets, covering different task (badminton and taxi) to different domain datasets (Rome and T-Drive), which further tests robustness under data drift.

**(2) Geometry Preserving Representation Learning**: We present a representation learning framework that unifies triplet objectives with loss design while preserving spatial and temporal geometry. The method anchors the distances between the positive and negative sample, combine with DMT loss to learn key information and structure from the negative similar and the most similar targets. We apply temporal embedding distance to preserve the desired pairwise temporal relations. To our knowledge we are the first to preserve the geometry of the original data features within the embedding space during training. Our method achieves 34.8% and 72.4% reduction in total training time compared to SOTA on Badminton and T-Drive. The average query latency is 8.3% lower than SOTA and remains far below the original TP. Across three datasets our method improves over the SOTA by 127.8%, 11.2%, and 10.9%. All results are obtained on the same device, and details are provided in Appendix D.6. After spatiotemporal convergence the learned representations yield clusters and analyses with meaningful class separation and stronger representativeness (Fig. 3 and Fig. 4).

**(3) Sequence Aware Spatiotemporal Integration**: Most existing trajectory models consider spatial features and sometimes temporal attributes, but rarely account for sequence dependencies, limiting their applicability to event-driven sports trajectories. Badminton movement players often return to center position, producing fragmented trajectory sequences (Wang et al. (2023a) and Fig. 7). As a result, models relying solely on average pooling or linear attention tend to miss cross-segment dependencies. To address discrete sports actions, NTAP captures higher-order feature interactions through a nonlinear projection, and employs a learnable context vector to dynamically reweight different segments. Through this sequence-aware integration, event-driven sports trajectories can achieve representation quality comparable to continuous GPS trajectories. The method overcomes the earlier focus on smooth continuous trajectory models, and achieves strong results on Badminton data, thereby filling an important gap in the literature.

## 2  PRELIMINARIES

**Unified Definition.** We define a trajectory $\mathcal{T} = \langle s_1, s_2, \ldots, s_n \rangle$ as a sequence of 2D locations over time. A sequence $s_i = (\mathrm{loc}_i, t_i)$ includes the 2D location $\mathrm{loc}_i = (x_i, y_i)$ as the coordinates and the observed time $t_i$ of the location. We model the trajectory in a trajectory network as a directed graph $G = (L, E)$, where $L$ is the set of trajectory vertices and $E$ is the set of directed edges corresponding to feasible trajectory segments. Each vertex $\ell_i \in L$ denotes a location in which its coordinate is $(x_i, y_i)$. An edge $e_{ij} = (\ell_i, \ell_j) \in E$ models a trajectory segment from $\ell_i$ to $\ell_j$.

**Trajectory Characteristics.** Different trajectory sources lie along a spectrum of properties and aren't limited, as the examples shown below:

- **Urban Mobility**: constrained by spatial topology, continuous, low tortuosity.
- **Basketball or Soccer**: no spatial topology constraints, continuous, moderate tortuosity.
- **Badminton or Tennis**: no spatial topology constraints, discrete, moderate–high tortuosity.

We define the characteristics of a **discrete trajectory** are being (1) event-triggered, (2) returning to a reset position (Wang et al., 2023a), and (3) following an episodic temporal structure. In contrast,

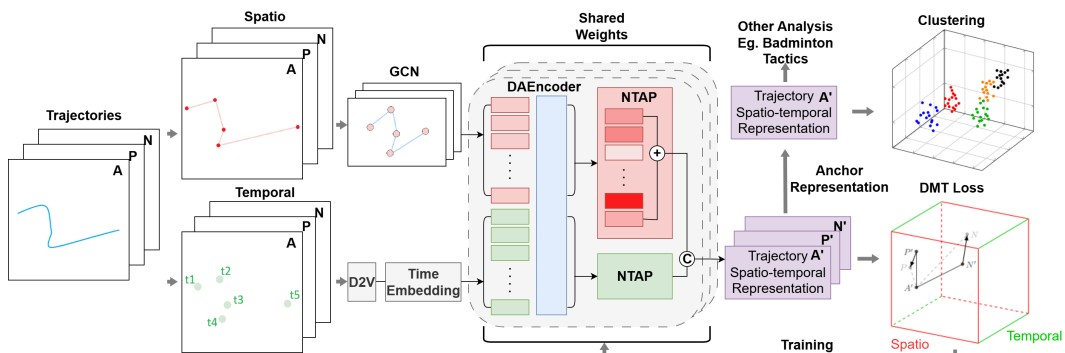

Figure 2: Overview of TRIDENT framework.

the characteristics of a **continuous trajectory** are (1) continuous, (2) no-reset, and (3) constantly adjusted behavior. Furthermore, we utilize $Average\ CCR$ defined in Yu et al. (2025) to quantify tortuosity. A higher $Average\ CCR$ indicates greater tortuosity. In our datasets, urban mobility exhibits relatively low tortuosity, whereas badminton exhibits high tortuosity (see Appendix D.2 for details). The distinction between different trajectories is non-binary and arises from their inherent behavioral organization and control mechanisms, rather than from differences in sampling rate. Therefore, we choose urban mobility and badminton as two extreme cases.

**Urban Mobility.** For urban mobility, a taxi with GPS as an example. GPS records latitude and longitude over time, where $x_i$ is the latitude, $y_i$ is the longitude, and $t_i$ is the timestamp of the GPS reading.

**Badminton Movement.** The half-court layout is shown in Fig. 7, as it is the range of movement of the player instead of the full court $(6.1\,\mathrm{m} \times 13.4\,\mathrm{m})$. Each half-court measures $6.1\,\mathrm{m} \times 6.7\,\mathrm{m}$, and the spatial unit is $0.1\,\mathrm{m}$. The coordinate origin $(0, 0)$ at the lower left corner and the axis extending to $(61, 0)$ and $(0, 67)$. To avoid duplicate patterns in different absolute locations, for any location $(x, y)$ with $y > 67$, the mapped location is $(2 \cdot 30.5 - x,\ 2 \cdot 67 - y)$. Player movements are captured as a stroke-by-stroke sequence, ordered by the stroke index within each rally. As $(x_i, y_i)$ is the location of the half court where the player plays the stroke on the stroke index $t_i$. Additionally, we sequentially index all integer grid coordinates in the standardized half court as vertices $(61 \times 67$ vertices). Therefore, the vertex is instead $\ell_i = \pi(\mathrm{loc}_i)$, where $\pi : \mathbb{R}^2 \to L$ denote the nearest-vertex projection.

**Learning Target.** Before learning, we need to choose the similarity measure as a learning target. There are many measures available, such as Yi et al. (1999) for spatial metrics, TP Shang et al. (2017), DITA Shang et al. (2018), and LCRS Yuan & Li (2019) for more advanced trajectory-based metrics. In this study, we chose TP as the main similarity measure. The reasons are: **(1) Less Emphasis on Sequence Order**, **(2) Considers Both Spatial and Temporal Dimensions**, and **(3) Nearest Neighbor Matching** (see Appendix D.5 for details).

## 3 METHOD

Our method consists of (1) a hyperparameter-free training strategy for constructing informative triplets, (2) a Distance-preserving Multi-kernel Triplet Loss (DMT) that aligns embedding-space distances with native trajectory-space distances across multiple scales, and (3) a Dual-Attention Encoder (DAEncoder) equipped with Nonlinear Tanh-Projection Attention Pooling (NTAP) for modeling both continuous and discrete trajectories.

Three limitations motivate our framework: (1) classic contrastive and triplet losses enforce relative ranking but not absolute geometric constraints, often causing dimension or class collapse (Jing et al., 2022; Xuan et al., 2020); (2) classic contrastive and triplet losses rely on sensitive hyperparameters like margins, temperatures, positive/negative ratios, and mining strategies (He et al., 2020; Schroff et al., 2015), triplet diversity or data augmentation (Chen et al., 2020), making training unstable and distorting the underlying geometry; (3) Previous trajectory representation methods handle continu-

ous trajectories well, but are less effective for discrete trajectories. Our design is guided by three principles:

- **Preserve intrinsic geometry:** Distances in the embedding space must respect distances in the native trajectory space at both local and global scales, preventing collapse.
- **Eliminate hyperparameter sensitivity:** Triplet construction and the loss should operate without margins, temperatures, or mining heuristics.
- **Maintain structural stability across trajectory types:** The system must handle continuous and discrete trajectories using the same architecture.

Below we describe the training strategy, loss formulation, and encoder architecture.

## 3.1 TRAINING STRATEGY

We leverage a similarity measure TP to construct triplets in a self-supervised fashion. For each anchor trajectory, we identify its most similar trajectory as the positive example. This captures the strongest local structural relationship while avoiding oversmoothing due to multiple positives. Negatives are drawn randomly from trajectories that are not the anchor nor its top-1 positive. This supplies unbiased global diversity without hard-negative mining.

This top-1 positive + random negative strategy reflects two motivations: (1) **local identity** is defined by the closest neighbor, and mutual top-1 relationships already imply a strong constraint without artificially compressing distances; (2) **global geometry** is learned through diverse negatives without requiring the user to specify a negative-to-positive ratio.

## 3.2 DISTANCE-PRESERVING MULTI-KERNEL TRIPLET LOSS(DMT)

A limitation of classic triplet loss even with margin is that it only enforces a relative ordering ($d'_{ap} < d'_{an}$), without constraining the magnitude of distances. This often distorts the embedding geometry, collapsing local neighborhoods, or exaggerating long-range separations, which degrades retrieval accuracy and cross-domain generalization. To mitigate this, we introduce the Distance-preserving Multi-kernel Triplet Loss (DMT), which explicitly aligns embedding space distances with ground truth native trajectory space distances, while weighting errors across multiple scales.

**Alignment.** Let $A', P', N' \in \mathbb{R}^{b \times d_{dim}}$ denote the predicted representations of anchor, positive, and negative samples of batch size $b$. Let $d'_{ap}, d'_{an} \in \mathbb{R}^b$ be the predicted embedding space distances and $d_{ap}, d_{an} \in \mathbb{R}^b$ be the ground truth trajectory-space distances. We compute the embedding and trajectory space distances as:

$$d'_{ap} = \|A' - P'\|_2, \quad d'_{an} = \|A' - N'\|_2. \quad d' = \begin{bmatrix} d'_{ap} \\ d'_{an} \end{bmatrix}, \quad d = \begin{bmatrix} d_{ap} \\ d_{an} \end{bmatrix} \in \mathbb{R}^{2b}. \quad (1)$$

The objective is to align embedding space distances with ground truth native trajectory space distances minimizing the squared error between $d'$ and $d$ as:

$$E^2 = (d' - d)^2 \in \mathbb{R}^{2b \times 1}. \quad (2)$$

However, by using native squared error, the error would be dominated by large-distance deviations, which fails to control local neighborhoods and does not retain global structure. Therefore, we apply a multi-scale Gaussian reweighting that normalizes error contribution by distance scale.

**Multi-kernel Gaussian weighting.** To compare distances across heterogeneous scales, we normalize the error constructing multiple Gaussian kernels. We first compute a base bandwidth $\sigma_{\text{base}}$ from the batch median ground-truth distance and multiply with fixed multipliers $m = [0.5, 1.0, 2.0]$ as $\kappa$ different scales of $\sigma$.

$$m = [m_1, \ldots, m_\kappa]^\top, \quad \sigma_{\text{base}} = \text{median}(d), \quad \sigma = \sigma_{\text{base}} \cdot m \in \mathbb{R}^\kappa. \quad (3)$$

Small $\sigma_k$ emphasizes accurate matching for small native distances (local neighborhoods), while larger $\sigma_k$ spreads attention to larger distances (global structure). Each kernel defines weights:

$$W = \exp\left(-\frac{d^2}{2\sigma^2 + \epsilon}\right) \in \mathbb{R}^{2b \times \kappa}. \quad (4)$$

**Scale-normalized residual.** For each scale $k$, we weight the squared errors $\boldsymbol{E}^2$ with $\boldsymbol{W}$ as:

$$\boldsymbol{\Delta} = \boldsymbol{W} \odot \boldsymbol{E}^2 \in \mathbb{R}^{2b \times \kappa}. \tag{5}$$

Then the scale specific loss is normalized as:

$$\ell_k = \frac{\mathbf{1}^\top \boldsymbol{\Delta}_{:,k}}{\mathbf{1}^\top \boldsymbol{W}_{:,k} + \epsilon} \in \mathbb{R}, \quad k = 1, \ldots, \kappa, \tag{6}$$

so that each kernel only penalizes deviations meaningful to its own distance regime. This prevents small-scale kernels from being dominated by large distances and prevents large-scale kernels from overweighting tiny local perturbations to further improve training stability.

**DMT loss.** Finally, the scale specific loss averages across scales:

$$\mathcal{L}_{dmt} = \frac{1}{\kappa} \sum_{k=1}^{\kappa} \ell_k \in \mathbb{R} \tag{7}$$

## 3.3 Model Structure

Each trajectory is represented by a spatial embedding (Appendix C.1) and a temporal embedding (Appendix C.2). We propose Dual-Attention Encoder, which employs a Nonlinear Tanh-Projection Attention Pooling mechanism to integrate these spatial and temporal representations, producing a unified representation architecture for continuous and discrete trajectories.

**Dual-Attention Encoder (DAEncoder).** DAEncoder adopts a post schedule with two classic cross attention in parallel. Within each layer the spatial embedding attends to the temporal, then the temporal embedding attends to the updated spatial. The two directions use independent parameters. Variable lengths and missing steps are handled with a learned CLS token for each representation, padding masks derived from true lengths, and zeroing of padded positions after positional encoding. Each embedding later is summarized by Nonlinear Tanh-Projection Attention Pooling (NTAP), and the two summaries are concatenated to form the final representation. We do not employ gating or add, so that spatial and temporal information contribute equally. The design is extensible to new attribute branches and multi-subject trajectories that follow the same fusion and pooling pipeline.

**Nonlinear Tanh-Projection Attention Pooling (NTAP).** To obtain fixed-size representations from variable-length spatial or temporal sequences, we employ a Nonlinear Tanh-Projection Attention Pooling mechanism. Traditional masked attention pooling typically relies on a single linear projection, which limits its capacity to capture higher-order feature interactions. In contrast, our design introduces a non-linear transformation prior to computing the attention scores, leading to more expressive sequence representations for discrete trajectories and retain the same performance on continuous trajectories.

Let the input sequence be $Z = \langle \boldsymbol{z}_1, \boldsymbol{z}_2, \ldots, \boldsymbol{z}_n \rangle$, $\boldsymbol{z}_i \in \mathbb{R}^{d_{dim}}$. First, each feature vector is first projected by a learnable matrix and passed through a $\tanh$ nonlinearity to capture nonlinear and higher-order interactions among feature dimensions, which are crucial for trajectory spatial and temporal data where dependencies are complex and dynamic, as

$$\boldsymbol{u}_i = \tanh(\boldsymbol{W}_\omega^\top \boldsymbol{z}_i) \in \mathbb{R}^{d_{dim}}, \quad \boldsymbol{W}_\omega \in \mathbb{R}^{d_{dim} \times d_{dim}}. \tag{8}$$

The nonlinear projection $\boldsymbol{u}_i$ preserves smooth variations, curved motion from continuous trajectories, and highlights sharp turns, high-curvature event points from discrete trajectories.

Second, a learnable context vector $\boldsymbol{u}_\omega \in \mathbb{R}^{d_{dim}}$ captures the perspective from which the sequence should be summarized, models nonlinear interactions across dimensions to emphasize key spatial regions in the trajectory and prioritize critical time steps. Event-triggered points and return points in discrete movements are more easily emphasized. Third, with a mask score to handle variable sequence lengths, where padding is masked by $mask_i$ as $-\infty$ and $0$ otherwise. Lastly, with softmax as the score

$$\alpha_i = softmax(\boldsymbol{u}_\omega^\top \boldsymbol{u}_i + mask_i) \in \mathbb{R}, \quad \tau \in \mathbb{R}, \quad mask_i \in \{0, -\infty\}. \tag{9}$$

Finally, the pooled representation that combined each feature vector according to the score $\alpha_i$ is calculated as

$$\boldsymbol{r} = \sum_{i=1}^{n} \alpha_i \boldsymbol{z}_i \in \mathbb{R}^{d_{dim}}. \tag{10}$$

# 4 EXPERIMENTS

## 4.1 EXPERIMENTS SETTING

We conduct experiments on the public Badminton, T-Drive, and Rome trajectory datasets. Parameter settings are in Appendix D.4, and full evaluation metrics are in Appendix D.4.1. Each dataset is split into 10,000 training trajectories, 4,000 for validation, and the remainder for testing. See Appendix D.1 for dataset details. Additionally, we perform experiments on two larger-scale datasets, with results available in Appendix D.7.

## 4.2 MODEL PERFORMANCE STUDY

Table 1: Framework effectiveness comparison on Badminton, T-Drive, Rome.

| Framework | Badminton | | | T-Drive | | | Rome | | |
|---|---|---|---|---|---|---|---|---|---|
| | HR@10 | HR@50 | R10@50 | HR@10 | HR@50 | R10@50 | HR@10 | HR@50 | R10@50 |
| NeuTraj (ICDE) | 0.009 | 0.036 | 0.041 | 0.177 | 0.222 | 0.270 | 0.123 | 0.217 | 0.261 |
| ST2Vec (KDD) | 0.073 | 0.178 | 0.212 | 0.496 | 0.593 | 0.844 | 0.479 | 0.598 | 0.830 |
| TrajCL (ICDE) | 0.084 | 0.243 | 0.197 | 0.241 | 0.369 | 0.359 | 0.094 | 0.293 | 0.260 |
| ConDTC (IEEE) | 0.009 | 0.021 | 0.047 | 0.139 | 0.348 | 0.343 | 0.091 | 0.279 | 0.323 |
| Our method | **0.190** | **0.347** | **0.484** | **0.564** | **0.661** | **0.916** | **0.535** | **0.664** | **0.913** |

To demonstrate the effectiveness of our model, we evaluate on three standard metrics, HR@10, HR@50, and R10@50, and compare against multiple baselines. As shown in Table 1, after averaging performance across these three metrics, our method exceeds the mean of all baselines by 271% on Badminton, 96% on T-Drive, and 127% on Rome. This demonstrates strong generalization across trajectories with diverse characteristics and highlights the stability of our approach. A notable observation is that ST2Vec and TrajCL achieve similar results on the Badminton dataset, suggesting that the contrastive positive and negative sampling strategy is more adaptable to complex and rapidly changing trajectories. In contrast, the absence of anchors causes CL to underperform relative to ST2Vec on the T-Drive and Rome datasets, which contain smoother trajectories, with a relative gap of 53.91%. Anchors are therefore crucial for continuous and smooth data, as also illustrated in Fig. 4.

## 4.3 MODEL EFFICIENCY STUDY

### 4.3.1 LEARNING PROCESS OF TRIDENT

To assess how the model learns trajectory representations, we used the converged model as a reference. We sampled 500 trajectories, computed their embeddings, clustered them via spectral clustering, and fixed the resulting labels. The same trajectories were then passed through models at three stages (Initial, Middle, Convergent), and their embeddings were projected to 2D with t-Distributed Stochastic Neighbor Embedding (t-SNE) to visualize how the cluster structure evolved during training. Fig. 3 shows embeddings progressing from densely overlapping clusters to partial separation and finally to clearly groups. Combining triplet loss with distance regression preserves the native spatial layout and pairwise distances. In the initial Fig. 3a, the pink and green groups are adjacent, as are the light and dark blue groups. After optimization, the convergent Fig. 3c maintains these local neighborhoods while cleanly separating distinct populations. In contrast, alternatives such as CL and ST2Vec tend to overfit, collapsing points into a single clump, which erases the original spatial structure.

### 4.3.2 ABLATION TESTS

Table 2a reports the best performing combinations of loss functions and our encoder. The higher capacity of the Transformer made it prone to unstable or suboptimal convergence with limited data. The GRU underfit the complex temporal and spatial patterns. Despite reasonable results on a small and complex dataset, the BiLSTM proved insufficient. Across loss functions combined with our DAEncoder, it yielded the most stable training dynamics, and the design of the loss function was

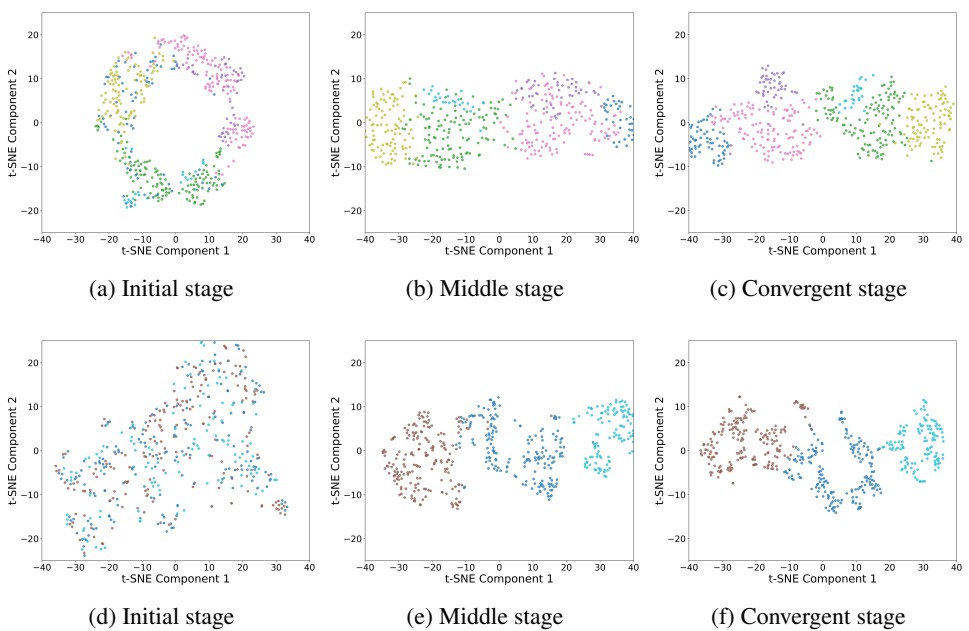

Figure 3: Visualization of learning process on Badminton (a–c) and T-Drive (d–f).

Table 2: Left: loss and encoder combinations. Right: attention pooling ablation.

(a) Loss × Encoder

T-Drive dataset

| Loss | Encoder | HR@10 | HR@50 | R@10 |
|------|---------|-------|-------|------|
| DMT | Transformer | 0.5038 | 0.6105 | 0.8688 |
| DMT | BiLSTM | 0.5168 | 0.6118 | 0.8736 |
| DMT | GRU | 0.4044 | 0.5298 | 0.7917 |
| DMT | DAE | **0.5643** | **0.6610** | **0.9160** |
| Pairwise Logistic CL | DAE | 0.5481 | 0.6319 | 0.9057 |
| Siamese Contrastive Loss | DAE | 0.2201 | 0.2487 | 0.4736 |
| RBF-Triplet Loss | DAE | 0.5094 | 0.6061 | 0.8688 |

(b) Attention pooling

T-Drive dataset

| Pooling | HR@10 | HR@50 | R@10 |
|---------|-------|-------|------|
| None | 0.5540 | 0.6539 | 0.9114 |
| Std-AP | 0.5506 | 0.6502 | 0.9038 |
| NTAP | **0.5643** | **0.6610** | **0.9160** |

Badminton dataset

| Pooling | HR@10 | HR@50 | R@10 |
|---------|-------|-------|------|
| None | 0.1477 | 0.2898 | 0.4014 |
| Std-AP | 0.1444 | 0.2805 | 0.3873 |
| NTAP | **0.1978** | **0.3596** | **0.5047** |

also critical. Table 2b compares models without attention. Simple pooling did not capture the salient features of the data. Our NTAP achieved the best trajectory modeling performance.

### 4.3.3 STABLE TESTS

Table 3: Generalization and data-efficiency evaluation (T-Drive).

(a) Cross domain

| Train | Test | Model | HR@10 | HR@50 | R@10 |
|-------|------|-------|-------|-------|------|
| T-Drive | Rome | TRIDENT | 0.1583 | 0.1934 | 0.3359 |
| Rome | T-Drive | TRIDENT | 0.1642 | 0.1782 | 0.3557 |
| T-Drive | Rome | ST2V | 0.0019 | 0.0097 | 0.0099 |
| Rome | T-Drive | ST2V | 0.1369 | 0.1638 | 0.3101 |

(b) Robustness to data diversity

| Reduction | HR@10 | HR@50 | R@10 |
|-----------|-------|-------|------|
| 0% | 0.5643 | 0.6610 | 0.9160 |
| 10% | 0.5585 | 0.6557 | 0.9031 |
| 30% | 0.5438 | 0.6502 | 0.9156 |
| 50% | 0.5195 | 0.6230 | 0.8937 |

Our method demonstrates strong architecture-general capability by achieving stable performance across three distinct types of datasets (Badminton, T-Drive, and Rome) using the same model architecture and identical hyperparameter settings. To further evaluate its cross-domain behavior, we additionally conduct transfer experiments across datasets: training on T-Drive and testing on Rome, and vice versa. The results show that our method outperforms baselines, and in cross-city transfer it

retains about 31% of its original performance. Although performance drops, this is mainly due to intrinsic differences in the data rather than limitations of the model. Cities differ substantially in GPS coordinate distributions, road topology, traffic rules, and movement constraints, making trajectory distributions impossible to align directly. Furthermore, GPS trajectories and badminton trajectories follow entirely different motion mechanisms. Their spatio-temporal dynamics do not overlap, and therefore badminton is not included in this test. Overall, these additional experiments demonstrate that our framework remains competitive even under large domain gaps. Rome has longer trajectories while T-Drive trajectories are shorter, making it difficult for existing methods to evaluate between them. In contrast, our model can differentiate them, even though it was never trained on this type of data.

## 4.4 CASE STUDY

### 4.4.1 THREE TYPE OF TRAJECTORY CLUSTERING

To evaluate our trajectory clustering, we visualize t-SNE on learned embeddings from the Badminton dataset. We apply spectral clustering (k=6) and color the clusters in the t-SNE plots. As shown in Fig. 4, our method preserves class cohesion and shows clearer inter-class separation than ST2Vec. ST2Vec embeddings collapse into two coarse lobes, causing overlap and poor within-class resolution. For TrajCL, adding temporal features further mixes trajectories into a single mass, likely because its loss lacks a triplet structure to anchor positives vs. negatives. Without an anchor that specifies which trajectories should be nearest and which should be farthest, the objective either (i) only pushes dissimilar pairs apart, or (ii) merely enforces higher similarity for positives than for negatives. Both cases fail to recover clean cluster boundaries. Preserving the intrinsic manifold structure also benefits downstream tasks: although our embedding appears more dispersed in t-SNE, this layout creates sufficient inter-cluster margins, clarifies temporal and spatial patterns, and enables more reliable clustering.

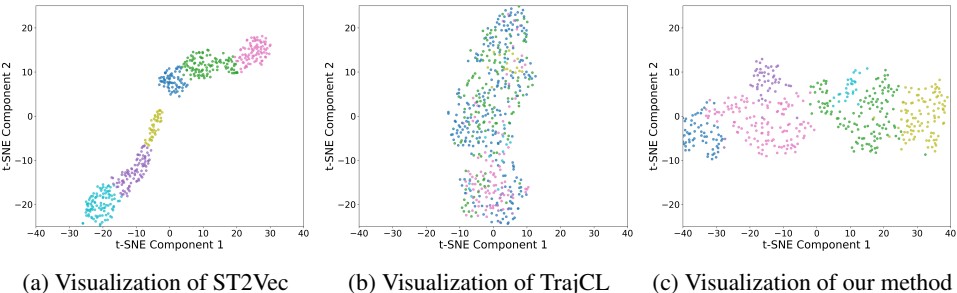

(a) Visualization of ST2Vec  (b) Visualization of TrajCL  (c) Visualization of our method

Figure 4: Visualization of Badminton dataset on different frameworks.

### 4.4.2 TOP-K T-DRIVE SIMILARITY QUERYING

We conducted top-k query experiments on the T-Drive dataset to evaluate the discriminative capability of our framework. A trajectory was randomly selected as the query trajectory. As shown in Fig. 5a, the top-2 trajectories identified by our method are highly consistent with the ground truth; in particular, Our retrieved top-1 and top-2 match the ground-truth order. In addition to this random case, we also include a deliberately chosen example with errors (Fig. 5b). Notably, in Fig. 5b, although the retrieved nearest neighbor to the ground truth (by distance) is closest numerically, the spatial pattern exhibits a hollow structure. Our method selects candidates based on global spatial consistency, thereby retrieving trajectories that are globally closer to the query.

### 4.4.3 K CLUSTER STRATEGY

We analyze only winner-ended rallies decided by a clean winner, excluding out, net contact, and other unforced errors to isolate tactics and positioning. Using the embedding from TRIDENT, we apply spectral clustering with k = 6 (Appendix D.9). This self-supervised pipeline partitions complex trajectories into clusters and retrieves instances with closely matched temporal order and spatial

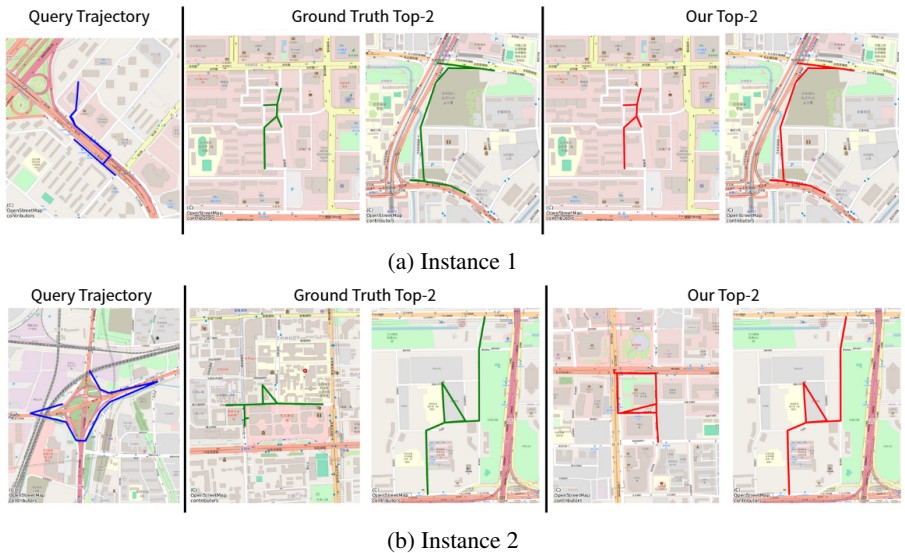

(a) Instance 1

(b) Instance 2

Figure 5: Top-k similarity querying (T-Drive).

positions, validating model effectiveness. The main text presents one representative cluster, with others in Appendix D.10. The cluster shows a defensive failure pattern. The typical sequence is opponent net shot, net shot, opponent lift, smash, opponent net shot, losing point. We term it the net lift smash cycle. Fig. 6 shows the three trajectories closest to the medoid, with numeric labels indicating movement order. Repeated front-to-back transitions let the attacker maintain initiative and disadvantage the defender. The recurrence of this sequence near the medoid indicates an actually executed tactic rather than a clustering artifact. The result aligns with the front-to-back pressure principle, provides quantitative support for coaching knowledge, and demonstrates how the model links unlabeled trajectories to interpretable tactics, forming a discovery-to-verification loop. In contrast, naive clustering of raw time and coordinate data (or baselines lacking trajectory awareness) cannot surface patterns that simultaneously capture temporal order and continuous spatial movement. Our approach can.

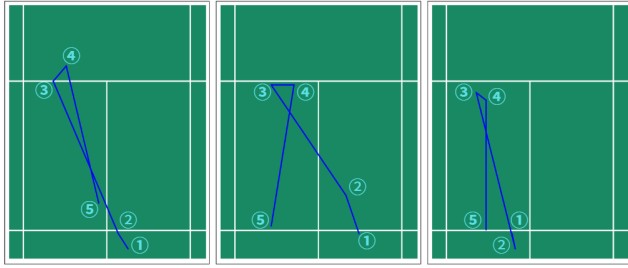

Figure 6: Badminton strategy.

## 5 CONCLUSION

This work presents a deep, self-supervised framework with a near–zero-hyperparameter design that preserves spatial, temporal, and sequential structure during representation learning and clustering. We replace ranking losses with distance regression to align embedding distances with real-world proximity, retaining metric interpretability under compression. The model stabilizes each instance using its most informative neighbors and delineates cluster boundaries by tracking negatives. The approach captures sequential patterns in spatio-temporal data (shown on badminton singles trajectories). Future work includes cross-task evaluations (e.g., pedestrian trajectories, doubles rallies, basketball) and deeper interpretability.

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

## A    APPENDIX

### A.1    BANDMINTON TRAJECTORY

Fig. 7 depict badminton singles player movement trajectories within three shots. The coordinate system follows the same scale (1 meters per unit) and represents a badminton half-court (6.1 × 6.7m). To extract more concentrated trajectories for better readability, we filtered specific shot sequences (Net shot → Clear → Smash) to ensure a structured pattern. We identified the most densely clustered location for step 1 and sampled trajectories within a 5-unit radius of that point, selecting ten representative trajectories. The magenta lines indicate the striking player's movement.

As shown in Fig. 8, illustrates that the badminton counter strategy can be traced when the hitting location is restricted to a limited area and the shot type remains the same. Under these conditions, there are identifiable patterns that can be observed in the data features. The most challenging aspect in this field has been the lack of an applicable model, and until recently, research focusing on mining strategies in the sports domain has been rare. The vast range of possibilities and uncertainties significantly increases the difficulty of exploration

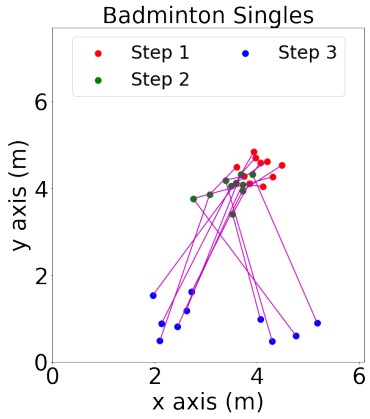

Figure 7: Movement patterns of consistent shot types and starting points in singles.

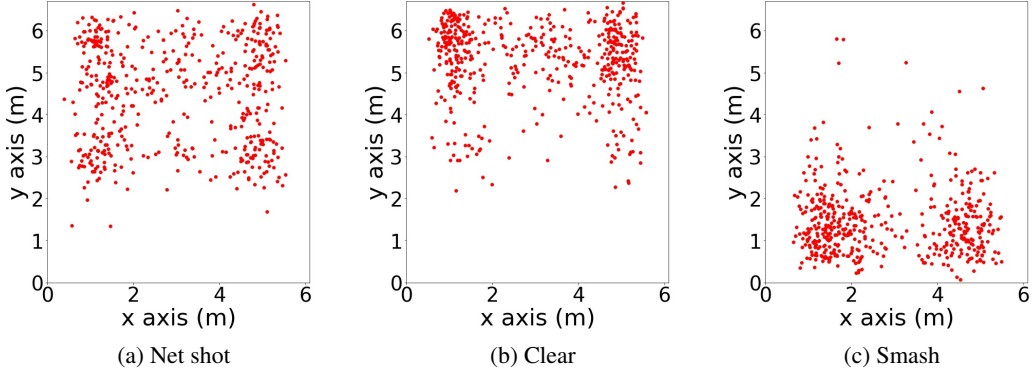

(a) Net shot     (b) Clear     (c) Smash

Figure 8: Shot hit distribution in badminton singles.

## B    RELATED WORK

### B.1    TRAJECTORY REPRESENTATION LEARNING

Trajectory clustering have introduced various techniques to enhance spatio-temporal pattern recognition. Deep embedding methods that preserve temporal dependencies Yao et al. (2019) work well

in smooth motion, but their accuracy degrades under irregular sampling, event-driven annotation, abrupt direction changes, and spatial attributes are usually not encoded jointly. Contrastive and end to end clustering approaches (Si et al., 2024; Fang et al., 2021a) improve discrimination, but they rely on sensitive augmentation choices and temperatures, and they optimize instance separation rather than calibrated cluster structure. Similarity oriented retrieval with learned metrics (Zhang et al., 2021) captures relationships defined by a preset notion of similarity, but temporal order is only weakly enforced and transfer across datasets is limited. Spatially constrained similarity Yang et al. (2021) strengthens spatial matching, yet fusion with temporal order is often late, which can bias the metric toward space. Trajectory contrastive learning Chang et al. (2023b) is sensitive to ordering and local window design, making long-range and multiscale dependencies difficult to learn in dynamic sports.

## B.2  TRAJECTORY CLUSTERING TECHNIQUES

In prior work, trajectory clustering has played a fundamental role in tasks such as trip classification (Lee et al., 2008), pattern detection (Chen et al., 2019). Its core objective is to group trajectories with similar movement characteristics so that those within the same cluster are mutually similar and clearly distinct from other clusters. Traditional approaches often use raw trajectory representations (paths or discrete points), define similarity measures such as the Euclidean or Hausdorff distance, and apply clustering algorithms(k-means). Gaffney et al. (2007) constructed an EM framework based on the global Euclidean distance between full trajectories. Wang et al. (2023c) employ a two-layer isolation distribution kernel (IDK) to represent trajectories and measure similarity. This approach achieved promising results on vehicle data, but failed to capture both the discrete attributes (e.g., shot type) and the continuous attributes (e.g., event location). Lee et al. (2007) segmented trajectories before clustering with the Hausdorff distance. However, such methods are sensitive to differing sampling rates, trajectory lengths, and noise. it may overlook crucial local features.To overcome the limitations of raw representations, deep trajectory embedding techniques have emerged. Fang et al. (2021b) introduces an end-to-end deep clustering framework via self-training to address the reliance on handcrafted similarity measures and capture latent spatial dependencies. Wang et al. (2013) incorporated action labels (entry, exit, stop, move) into event detection. More recently, Yue et al. (2019) extracted points of interest (POIs) from trajectories for deep clustering. Although these deep methods reduce manual feature engineering, they still depend to some extent on predefined labels or segments.

## B.3  STRATEGY DISCOVERY IN TEAM SPORTS

This study focuses on temporal and spatial data, using trajectory clustering and mining to uncover previously undefined strategies. Knauf & Brefeld (2014) introduce a novel spatiotemporal kernel that clusters multiple player trajectories simultaneously. This method captures strategic deviations between two teams. Warnakulasuriya et al. (2015) combines handcrafted features with the trajectories of players on both teams to distinguish different characteristics associated with scoring opportunities. Decroos et al. (2018) applies a similar technique to hockey to identify attacking strategies that result in continuous scoring. Another line of work focuses on ball movement. Although this approach contributes to understanding overall team strategies, it struggles to reveal fine-grained player trajectories. Trajectory mining and high-frequency spatio-temporal pattern mining both emphasize frequent patterns, though frequency does not necessarily imply strategic value.

## C  INPUT EMBEDDING MODULE

### C.1  SPATIAL EMBEDDING WITH GCN

To capture both the structural information and the sequential nature of trajectories, a trajectory embedding module that integrates graph convolutional networks (GCN) with a dedicated PAD node mechanism is used. Given the trajectory network $G = (L, E)$ defined in Section 2, each vertex $\ell \in L$ is associated with a feature vector $x_\ell \in \mathbb{R}^{d_{\text{in}}}$. Using a GCN, the node features are encoded into the hidden space:

$$H = \text{GCN}(X, A) \in \mathbb{R}^{|L| \times d_h}, \tag{11}$$

where $X \in \mathbb{R}^{|L| \times d_{\text{in}}}$ is the input feature matrix, $A$ is the adjacency matrix, and $H$ is the resulting node embedding matrix.

When processing trajectories in batches, sequences of different lengths must be aligned. Conventional approaches pad shorter sequences with zero vectors, which makes it difficult to distinguish padding positions from real nodes in the embedding space. A *PAD node* to the node embedding matrix:

$$H' = \begin{bmatrix} H \, ; \, \mathbf{0} \end{bmatrix} \in \mathbb{R}^{(|L|+1) \times d_h}. \tag{12}$$

During batching, all padded positions are assigned to this PAD node index, ensuring they are semantically separated from real nodes.

For a trajectory $\mathcal{T} = \langle s_1, s_2, \ldots, s_n \rangle$, its embedding sequence can then be obtained via direct indexing:

$$E_{\mathcal{T}} = [\, H'_{s_1}, H'_{s_2}, \ldots, H'_{s_n} \,] \in \mathbb{R}^{n \times d_h}. \tag{13}$$

GCN encodes structural information from the trajectory network into node embeddings and the PAD node ensures that padding positions are clearly distinguished from real nodes, thereby enabling efficient batch computation without compromising semantic correctness.

### C.2 TIME EMBEDDING

Temporal information is indispensable in trajectory modeling. A simple linear projection of temporal features into the hidden space often fails to preserve fine-grained cues, such as hour-of-day, weekday, or periodic patterns. Although these raw signals are simple, they frequently carry decisive semantic meaning. To address this issue, we design a time embedding module with a residual pathway, so that the model can learn high-level abstractions while progressively integrating raw temporal signals. Let each sampling point $s_i = (\text{loc}_i, t_i)$ consist of its location and timestamp, where $t_i \in \mathbb{R}^{d_{\text{in}}}$ denotes the temporal feature vector (e.g., date2vec encoding of the raw strole index or GPS time). We first apply a learnable linear projection:

$$z_i = W_p t_i + b_p, \qquad z_i \in \mathbb{R}^{d_h}, \tag{14}$$

where $W_p \in \mathbb{R}^{d_h \times d_{\text{in}}}$ and $b_p \in \mathbb{R}^{d_h}$ are learnable parameters. At the same time, maintains a residual branch directly from the raw temporal input:

$$r_i = W_s t_i + b_s, \tag{15}$$

with $W_s \in \mathbb{R}^{d_h \times d_{\text{in}}}$ and $b_s \in \mathbb{R}^{d_h}$ initialized to zero. This initialization ensures that during the early stages of training, the residual path has no effect, allowing the model to focus on learning stable projected representations. As training progresses, the parameters of the residual branch can gradually adjust, enabling the network to incorporate raw temporal cues when necessary. The final temporal embedding for step $i$ is obtained as

$$h_{t_i} = \text{Dropout}\big(\sigma(z_i + r_i)\big), \tag{16}$$

where $\sigma(\cdot)$ denotes the ReLU activation. Through this design, the embedding is able to capture high-level abstractions without discarding the influence of low-level raw features. When large-scale data is available, the model benefits from learning complex temporal dynamics through the projection branch; when data is scarce, the residual path ensures direct access to raw signals, thereby improving generalization and robustness. Preserve raw temporal cues while progressively integrating them into higher-level abstractions, ensuring stable and expressive temporal representations across data regimes.

## D EXPERIMENTS

### D.1 DATASET

- Badminton (Wang et al., 2023a;b) : Movement trajectories from singles matches. After handling missing data and removing rallies with <5 strokes, 12,667 rallies remain. Each trajectory uses the last five strokes, considering both players yields 25,334 trajectories. The test set contains 11,334 trajectories.

- T-Drive (Yuan et al., 2010; 2011) : The dataset contains approximately 15 million taxi GPS points collected from Feb 2 to Feb 8, 2008. To enable road network-aware trajectory similarity analysis, raw GPS traces are map-matched to OpenStreetMap and converted into time-ordered sequences of road vertices. Restricting to urban trips and filtering trajectories with fewer than 10 samples leaves 29,830 trajectories. The test set contains 15,830 trajectories.

- Rome (Bracciale et al., 2022) : The dataset comprises 30,000 taxi trajectories from Rome, Italy, spanning over 30 days. Following the same map matching and filtering procedures as applied to T-Drive. The test set contains 16,000 trajectories.

To enhance the completeness and reliability of our experiments, we incorporated two additional large-scale datasets from different cities. We conducted further training and evaluation. The details and experimental results of the added datasets are presented below and further in Appendix D.7.

- Chengdu (DiDi Chuxing, 2022) : This dataset collects taxi GPS trajectories in Chengdu, China, during October and November 2016. The raw GPS traces are map-matched to the OpenStreetMap network and converted into time-ordered sequences of road segments. The dataset contains a total of 84,251 trajectories, which are partitioned into a training set of 20,000, a validation set of 15,000, and a test set of 49,251 trajectories.

- Xian (DiDi Chuxing, 2022) : This dataset comprises trajectories from Xian, China, spanning the same period. Following the map-matching process to align traces with the road network, the dataset consists of 199,999 trajectories. We utilize 60,000 trajectories for training, 20,000 for validation, and the remaining 119,999 trajectories for testing.

## D.2 DATASET GEOMETRY ANALYSIS

Table 4: Dataset CCR statistics.

| Dataset | Count | Mean | Median | Std | Max |
|---|---|---|---|---|---|
| Badminton | 24036 | 4.3894 | 3.2325 | 3.2921 | 17.4368 |
| T-Drive | 31681 | 1.6730 | 1.3136 | 1.0681 | 8.1906 |

Besides the mentioned discrete and continuous properties, we explain here the geometric differences between the badminton and taxi trajectory datasets by analyzing Curvature Compression Ratio (CCR) from Yu et al. (2025), defined as the ratio of the trajectory arc length to the straight-line distance between its start and end points. Since both datasets occasionally contain extreme CCR values due to noise such as identical starting points, we apply a quantile 0.95 to 1 truncation strategy, removing the top 5% of outliers to preserve the intrinsic distribution of the data. The metric we use is:

$$Average\ CCR = \frac{1}{N} \sum_{i=1}^{N} \frac{L_i}{D_i} \tag{17}$$

where $L_i$ is the length of the $i$-th segment, $D_i$ is the distance between its start and end points, and $N$ is the total number of segments.

- When $Average\ CCR = 1$, the trajectory is perfectly straight.

- When $Average\ CCR > 1$, the trajectory is more curved and deviates more from a straight line; a larger value indicates higher tortuosity.

In Fig. 9, the badminton trajectories show a significantly higher Average CCR in the histogram, with a longer tail to the right overall, reflecting higher curvature and greater diversity. In contrast, T-Drive trajectories have an mean Average CCR around 1.9, consistent with the smoother movement patterns expected in continuous trajectories. This geometric difference supports the need to consider both high-tortuosity and low-tortuosity trajectory types when evaluating models.

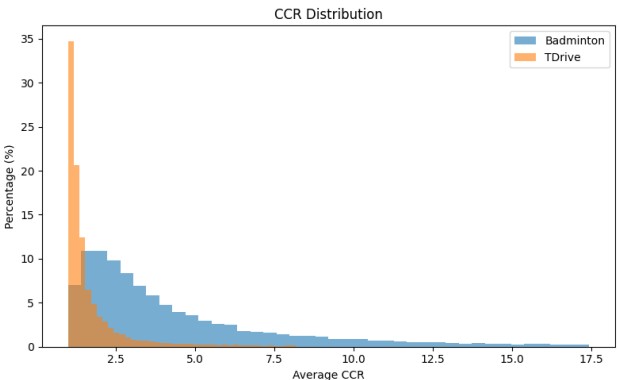

Figure 9: CCR distributions of Badminton and T-Drive.

## D.3 BASELINE

To comprehensively evaluate the effectiveness of our method in trajectory representation, we compare it against four representative models: ConDTC, ST2Vec, NeuTraj, and TrajCL. These methods cover the major technical directions in current trajectory research:

- ConDTC: Combines a Transformer encoder with location and temporal embeddings, together with contrastive clustering, for fine-grained spatio-temporal pattern recognition.
- TrajCL: Utilizes multiple trajectory augmentation strategies, contrastive learning, and an improved dual-feature self-attention encoder, representing contrastive trajectory representation learning approaches.
- ST2Vec: Employs triplet loss and a spatio-temporal co-attention mechanism to emphasize temporal–spatial details in similarity computation, serving as a key method for time-sensitive similarity modeling.
- NeuTraj: Uses an RNN backbone with a distance-weighted ranking loss to preserve similarity order, and is often applied for approximating high-cost similarity measures such as Frechet and DTW.

These four models span different architectural paradigms (Transformer-based, self-attention–based, and RNN-based encoders) and incorporate a range of loss functions, including contrastive, triplet, ranking, and distance-weighted ranking losses. Together, they form a representative and balanced set of baselines for learning trajectory representation.

## D.4 PARAMETER SETTINGS

In Our framework, the parameters are set with feature_size and embedding_size = 64. Batch size, and learning rate are configured as 128, and 0.001. Training and evaluation processes were conducted on a system running Python 3.10.18 ,PyTorch 2.5.1 with cu121. Implementation was conducted in Tensorflow on a system with an Intel(R) Core(TM) i9-14900KF CPU with an Nvidia RTX 4090 GPU. Choosing embedding_size 64 avoids giving our method an unfair advantage from a larger latent space and keeps the computational cost low. For batch size and learning rate, we adopted the values that produced the most stable training behavior on the T-Drive validation dataset. Batch sizes of 32, 64, and 128 reached nearly identical performance levels toward the end of the HR@10 training stage. Therefore, we selected a batch size of 128 to maximize training efficiency. Similarly, learning rates of 0.0005 (red line) and 0.001 (green line) both converged to comparable performance, indicating that either value would be suitable. We simply chose 0.001 as the learning rate.

### D.4.1 EVALUATION METRICS

To evaluate the effectiveness of trajectory similarity learning, we adopt the top-k similarity search framework, following previous studies in this area (Fang et al., 2022). Specifically, we report on

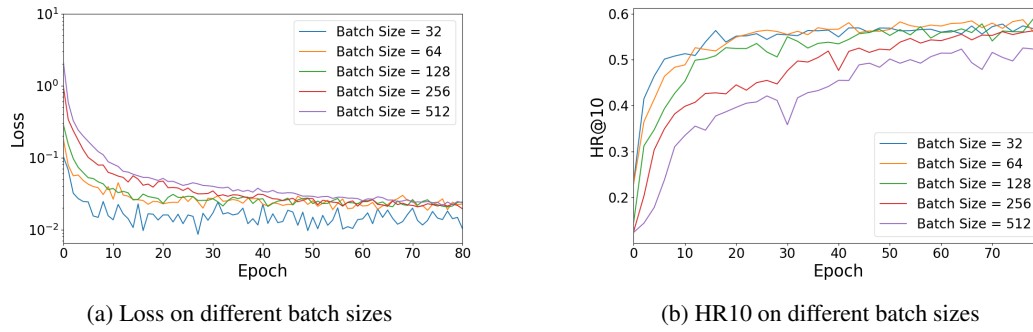

(a) Loss on different batch sizes
(b) HR10 on different batch sizes

Figure 10: Effects of different batch sizes.

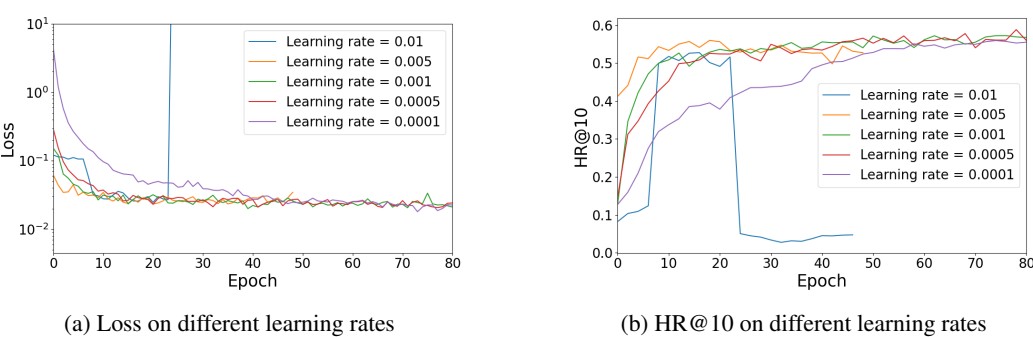

(a) Loss on different learning rates
(b) HR@10 on different learning rates

Figure 11: Effects of different learning rates.

three commonly used evaluation metrics: HR@10, HR@50, and R10@50. HR@k (Hit Rate at k) measures the proportion of queries for which at least one ground-truth trajectory appears in the top-k results returned by the learning-based method. Reflects the accuracy of the retrieval at a fixed cut-off. Rk@t (Recall at t for top-k ground truth) evaluates the fraction of the top-k ground-truth trajectories that are retrieved within the top-t candidates by the model. A higher value indicates better retrieval performance and more effective similarity preservation in the learned embedding space. Ground-truth top-k similar trajectories are obtained using the TP (Trajectory Point) distance, and details are provided in Appendix D.5.

## D.5  GROUND TRUTH

We construct ground-truth top-$k$ neighbors using the **TP**. For a trajectory network $G = (L, E)$, let $d_G : L \times L \to \mathbb{R}_{\geq 0} \cup \{\bot\}$ be the shortest-path distance on $G$, where $\bot$ denotes "unreachable". A trajectory is a sequence of sampled nodes (optionally with timestamps): $T_i = \langle (\ell_k^i, t_k^i) \rangle_{k=1}^M$, $T_j = \langle (\ell_\ell^j, t_\ell^j) \rangle_{\ell=1}^N$.

The directed form is

$$\overrightarrow{S}_{\text{TP}}(T_i, T_j) = \max_{1 \leq k \leq M} \min_{1 \leq \ell \leq N} d_G(\ell_k^i, \ell_\ell^j). \tag{18}$$

The symmetric distance is the discrete Hausdorff form

$$S_{\text{TP}}(T_i, T_j) = \max\left\{ \overrightarrow{S}_{\text{TP}}(T_i, T_j), \ \overrightarrow{S}_{\text{TP}}(T_j, T_i) \right\}. \tag{19}$$

If there is no unreachable case in the spatial implementation, then TP is

$$\text{TP}(T_i, T_j) = \left\lfloor S_{\text{TP}}(T_i, T_j) \right\rfloor. \tag{20}$$

The advantages are as follows:

**(1)Less Emphasis on Sequence Order**: TP places less emphasis on strict sequence order. In badminton strategy mining, data resemble taxi-route patterns where multiple paths may lead to the same

goal; similarly, tactical applications can vary while the intent remains consistent. TP handles such non-sequential differences well.

**(2)Considers Both Spatial and Temporal Dimensions**: The TP metric considers both spatial and temporal similarity, without requiring exact sequence order. This allows the TP metric to adjust more flexibly when dealing with complex spatiotemporal data, making it more adaptable to real-world scenarios.

**(3)Nearest Neighbor Matching**: By aggregating via the maximum of nearest-neighbor distances, small local errors are tolerated but clear deviations are amplified, preserving both stability and separability without extra decay/alignment hyperparameters.

### D.6 LOSS AND ACCURACY

From the training curves, it is clear that we did not stop early. For comparability with SOTA, we truncate training at epoch 326 in Fig. 12a. In our approach, convergence typically occurs between 290 and 330 epochs on the Badminton dataset, while on the T-Drive dataset, convergence occurs between 100 and 130 epochs. In comparison, ST2Vec converges at epoch 137 on Badminton and epoch 98 on T-Drive on average. Different methods define their losses with different scales and semantics, so we refrain from making comparisons between methods on raw loss values.

All experiments were conducted under identical CPU/GPU environments to ensure fair comparison. Our method demonstrates superior computational efficiency across both datasets: on Badminton, it achieves 3.4 seconds per epoch compared to ST2Vec's 12.6 seconds, resulting in total training times of 17.8 minutes versus 27.3 minutes, respectively. This efficiency advantage is even more pronounced on the T-Drive dataset, where our method requires only 9.7 seconds per epoch against ST2Vec's 39.7 seconds, leading to total training times of 17.8 minutes compared to 64.6 minutes. Notably, our method maintains consistent training duration (about 18 minutes) across both datasets despite their different characteristics, highlighting the robustness of our computational approach (hardware and hyperparameter settings details in Sec. D.4).

We evaluated query efficiency on the Badminton validation set (4,000 trajectories) by measuring the time to compute similarity between query trajectories and all others in the dataset. Our method achieves 35.2 ms to query against all 4,000 trajectories. The SOTA method needs 38.4 ms in total, representing 8.3% higher computation time. In contrast, traditional TP similarity requires 18.9 minutes, demonstrating the significant advantage of embedding-based approaches for large-scale trajectory analysis.

As shown in Fig. 12, because our model jointly optimizes the temporal and spatial dimensions and balance both, accuracy at a small hit radius (HR@10) is not expected to be particularly high. However, since our model learns the spatiotemporal distance directly and minimizes localization error, the overall accuracy keeps improving over training and surpasses existing SOTA, and the advantage becomes more pronounced as the radius decreases.

### D.7 SUPPLEMENTARY EXPERIMENTS ON LARGE DATASETS

Table 5: Results on the large-scale dataset.

| Framework | Chengdu | | | Xian | | |
|---|---|---|---|---|---|---|
| | HR@10 | HR@50 | R10@50 | HR@10 | HR@50 | R10@50 |
| ST2Vec (KDD) | 0.5437 | 0.6166 | 0.8594 | 0.3944 | 0.3856 | 0.6823 |
| Our method | **0.6802** | **0.7681** | **0.9769** | **0.4647** | **0.5227** | **0.8108** |

To further validate the robustness and generalizability of our model, we additionally expanded the dataset. With the larger data set (21.1% in Chengdu, 24.1% in Xian), the performance improvements over the SOTA baselines are substantially greater than those observed in the original data set (11.2% in T-Drive, 10.9% in Rome). These results not only confirm the model's stability under data scaling but also demonstrate its strong ability to generalize across diverse domains.

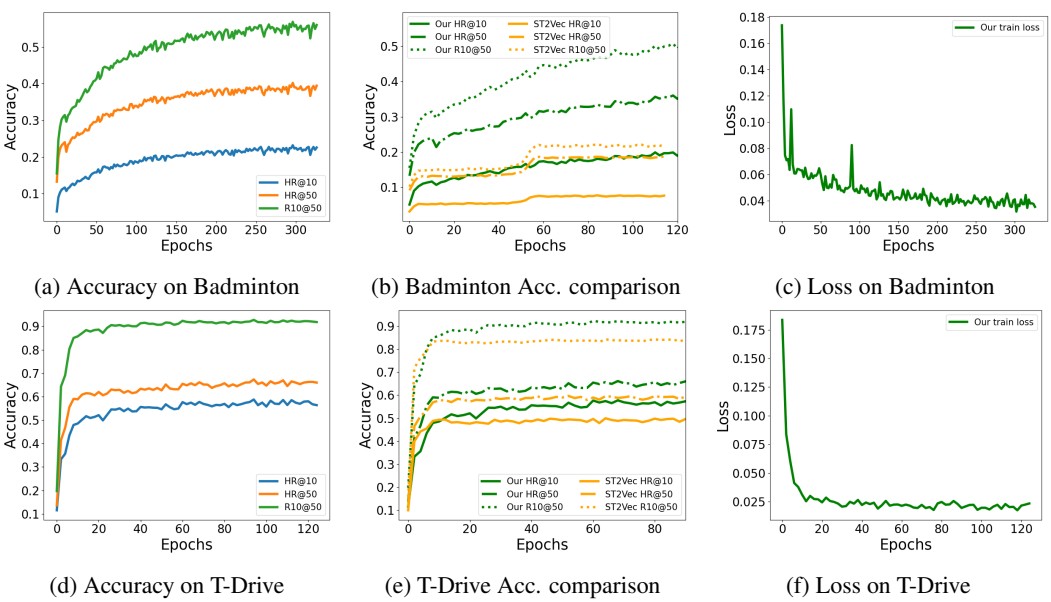

Figure 12: Visualization of accuracy and loss.

Table 6: Framework ranked effectiveness comparison on Badminton, T-Drive, Rome.

| Framework | Badminton | | | T-Drive | | | Rome | | |
|---|---|---|---|---|---|---|---|---|---|
| | HR@10 | HR@50 | R10@50 | HR@10 | HR@50 | R10@50 | HR@10 | HR@50 | R10@50 |
| NeuTraj | 4.5 | 5 | 5 | 4 | 5 | 5 | 3 | 5 | 4 |
| ST2Vec | 3 | 4 | 2 | 2 | 2 | 2 | 2 | 2 | 2 |
| TrajCL | 2 | 2 | 3 | 3 | 3 | 3 | 4 | 3 | 5 |
| ConDTC | 4.5 | 3 | 4 | 5 | 4 | 4 | 5 | 4 | 3 |
| Our method | **1** | **1** | **1** | **1** | **1** | **1** | **1** | **1** | **1** |

## D.8 FRAMEWORK SIGNIFICANCE TEST

**Friedman Test.** Table 6 shows that our model consistently achieves rank 1 across all metrics and datasets compared to various baselines. To assess the statistical significance of these improvements, we first conducted the Friedman test to determine whether there are significant differences among the evaluated frameworks. The test yields a p-value of $1.078 \times 10^{-5}$, which is well below the conventional threshold of 0.05, indicating the presence of statistically significant differences. Consequently, we further performed a Nemenyi post-hoc test to identify which frameworks differ significantly from our method.

Table 7: Nemenyi test (pairwise p-values).

| Model | NeuTraj | ST2Vec | TrajCL | ConDTC |
|---|---|---|---|---|
| NeuTraj | 1 | - | - | - |
| ST2Vec | 0.02999 | 1 | - | - |
| TrajCL | 0.33728 | 0.83507 | 1 | - |
| ConDTC | 0.97568 | 0.14139 | 0.71148 | 1 |
| Our method | 0.00002 | 0.38002 | 0.03727 | 0.0003 |

**Nemenyi Test.** Table 7 presents the pairwise p-values from the Nemenyi test based on ranks. The results indicate that our method significantly outperforms NeuTraj, TrajCL, and ConDTC (p-value $< 0.05$). While the comparison with ST2Vec shows a smaller level of significance, our method still achieves higher numerical performance and better ranks across all datasets. Notably, Table 1

demonstrates that our model outperforms ST2Vec on the badminton dataset, and Table 5 shows that the performance gap increases on larger datasets.

### D.9 CLUSTER NUMBER

To assess the influence of the number of clusters on segmentation quality, we conducted a series of parameter tests within the context of our proposed framework. This analysis was performed specifically on our curated Badminton dataset, which contains rich spatiotemporal patterns suitable for evaluating segmentation strategies.Clustering quality was assessed via the silhouette coefficient across $k \in \{3, \ldots, 9\}$; the best-scoring $k$ was used in subsequent experiments. As reported in Fig. 13, the results consistently indicate that setting n_cluster=6 yields the most favorable outcomes across all evaluation metrics. This suggests that a six-cluster configuration provides a well-balanced representation of underlying trajectory structures in badminton rallies, effectively capturing their tactical diversity while avoiding both over- and under-segmentation.

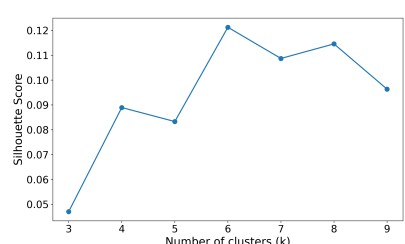

Figure 13: Silhouette scores on Badminton.

### D.10 CASE STUDY - BADMINTON STRATEGIES

The clustering analysis reveals six distinct tactical patterns, each characterized by specific shot sequences and strategic positioning implications. In Fig. 14, red points represent the starting positions of rally sequences, while yellow points indicate the ending positions where points are won or lost. Three distinct defensive failure patterns emerge from the analysis, each demonstrating how specific tactical weaknesses can be systematically exploited by opponents. In Fig. 14a, the **net lift smash cycle** reveals a defensive failure pattern where the sequence begins with opponent net shot, net shot, opponent lift, smash, opponent net shot, losing point. This pattern demonstrates how repeated front-to-back transitions allow the attacker to maintain initiative while placing the defender at a continuous disadvantage. The **flat defense collapse** in Fig. 14b follows a similar trajectory, beginning with opponent smash, flat, opponent lift, smash, opponent net shot, losing point. In this sequence, the defender's flat return fails to regain initiative, enabling the opponent to sustain pressure through alternating rear and front court attacks, which aligns with the principle that passive flat defense frequently leads to forced errors or delayed responses at the net. The third pattern in Fig. 14c, **passive reset after attack**, illustrates how conservative play can backfire even when initially positioned advantageously. The sequence progresses from opponent lift, smash, opponent flat, clear, opponent smash, losing point, where although the attacker initiates with a smash, the defensive flat shot neutralizes momentum, yet the subsequent clear cedes initiative, allowing the opponent to deliver a decisive counter-smash. These patterns collectively illustrate how front-to-back pressure principles can be weaponized against defenders who fail to break the cycle or regain tactical initiative.

In contrast to the defensive failures, three successful offensive strategies demonstrate how tactical sophistication and tempo control can generate winning opportunities. The **midcourt kill after rotation** strategy in Fig. 14d showcases how strategic positioning can create decisive openings, with the sequence unfolding as net shot, opponent lift, clear, opponent drop/smash, net shot, winning point. This pattern demonstrates that while both sides engage in front-back transitions, the eventual winner converts the rally through a midcourt kill opportunity, highlighting the tactical value of combining early net control with rear-court variation to open midcourt space. In Fig. 14e, the **tempo shift to rear attack** strategy emphasizes rhythm disruption as a pathway to offensive success, beginning with a drop, opponent push, flat, opponent clear, smash, winning point. Here, the attacker employs a drop to disrupt the rally's rhythm, forcing the opponent into a less aggressive push response and subsequently a clear, which provides the opportunity to seize initiative with a decisive rear-court smash. Finally, the **net exchange turnaround** strategy in Fig. 14f demonstrates the power of tactical recovery, where the sequence moves from clear, opponent smash, net shot, opponent net shot, lift/net shot, winning point. Despite facing initial defensive pressure, the attacker successfully balances the rally through strategic net exchanges and ultimately regains initiative, confirming how proactive net

play can transform defensive positions into offensive opportunities and supporting the fundamental principle of front-court dominance.

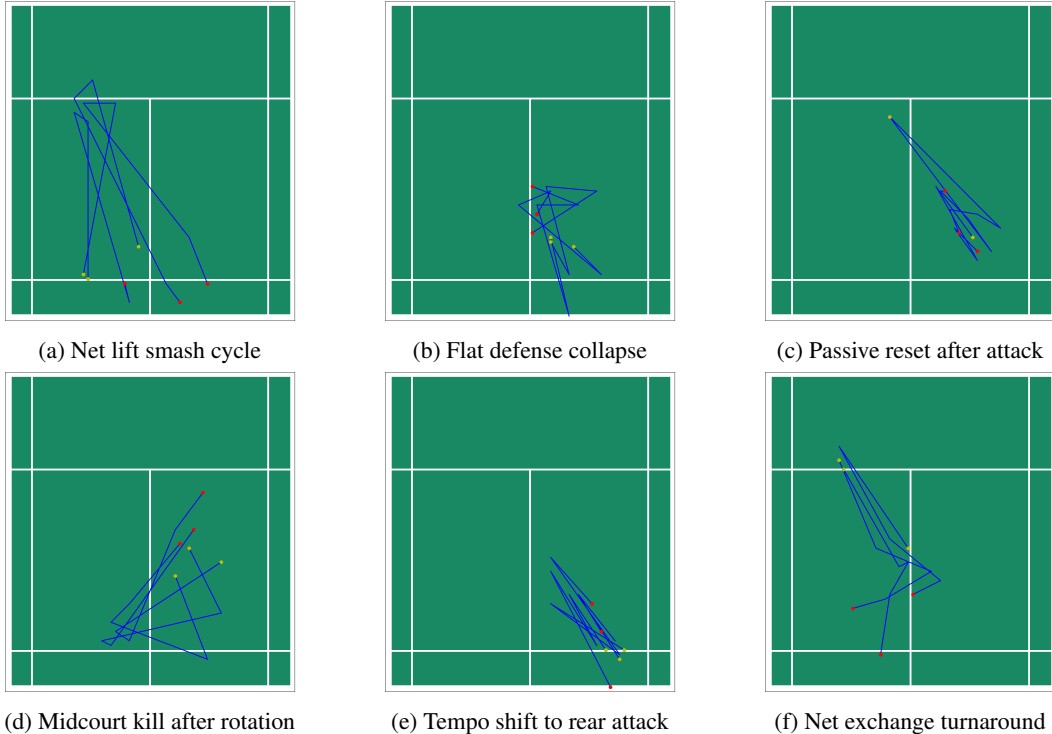

(a) Net lift smash cycle          (b) Flat defense collapse          (c) Passive reset after attack

(d) Midcourt kill after rotation   (e) Tempo shift to rear attack     (f) Net exchange turnaround

Figure 14: Visualization of badminton strategies.

