# OpenReview forum: "TRIDENT: Cross-Domain Trajectory Spatio-Temporal  Representation via Distance-Preserving Triplet Learning"
_ICLR.cc/2026/Conference — ICLR 2026 Poster_

### Official Review · Reviewer_d5b8 · 2025-10-28

**Soundness:** 4
**Presentation:** 2
**Contribution:** 3
**Rating:** 6
**Confidence:** 4

**Summary:**

The paper presents a spatio-temporal representation framework that can handle perturbations in trajectories instead of assuming smooth and continuous motion. It also presents a loss function that preserves local order and robustness under noise. The framework was tested against 4 baselines using 3 datasets from different domains and demonstrated improvements.

**Strengths:**

- The paper addressed a relevant problem regarding extracting trajectories for perturbation data, specifically in sports-related data.
- Very good introduction of the problem and the relevant research gap. The authors properly introduce the problem of not being able to simultaneously learn both point- and shape-based trajectories, in addition to the perturbation measures due to errors is measurement and sampling then highlight the particular challenges in sport trajectories and how they are relatively underexplored.
- A scalable and unified solution. The paper utilities a unified architecture that can be trained on different data types, thus avoiding redesigning distances and model components.
- Very good experiments and results. The authors conducted experiments spanning multiple trajectory types and showed improvements across all of them.

**Weaknesses:**

- Presentation issue (I recommend fixing these in case the paper proceeds to be accepted)
- Incorrect usage of “\citep” and “\citet” across the paper.
- The appendix section is not properly numbered. The appendix should have letter “A, B, …” instead of numbers.
Some acronyms are used without being presented or before being fully presented. Examples: NTAP, GCN,
- Very long introduction. The current introduction is very longs and includes both the introduction and the related work. It is better to have these sections separated for readability.
- Unclear setting of hyperparameters. The authors do not provide details on how the hyperparameter of the model are set, nor the reasoning behind setting these hyperparameters. There is only a brief description in line 774-779.
- The baseline methods are not properly presented. The authors compared their experiments with 4 baselines methods. However, the methods are not presented in details. It is important to present the details of these methods, regarding the architecture and the type of loss function they use. This assists in evaluating and understanding the experimental setup, in addition to clarifying the motivation of choosing these exact baselines.
- When comparing the framework presented in the paper with the baselines, the results are presented in numerical forms without testing whether the improvements are statistically significant or not. The statistical significance of the claimed improvements in comparison with the baselines methods needs to be tested (using a Friedman test followed by Nemenyi post-hoc test, for example).
- The code is not available for the review process. The authors mention in the paper abstract that “An anonymous repo with code and data is in the supplement.”. However, there is no such link in the appendix.

**Questions:**

- In the training strategy (lines 220–222), the positive and negative sample are selected based on calculating a similarity measure. For that, there are two considerations:
- Are there specific threshold to control these selections? If so, can you please elaborate on this? Without thresholds, it can happen that the positive sample is too close or identical to the anchor, and/or the negative sample is too far from the anchor. This hinders the effectiveness of contrastive triplet losses.
- Is the computation cost considered? And are there are pre-computing steps to ensure that the same calculation between the exact same sample is not done many times? This is important for the scalability of the proposed method.
- In lines 250–257, the authors discuss the values of k . However, there is no discussion nor mathematical notation showing how it is calculated. Can you please provide more details?
- In lines 773–779, a brief description of setting the hyperparameters is presented. However, only a few hyperparameters are presented without the reasoning behind setting these values. Can you please elaborate on why setting these exact values and provide details and reasoning about other hyperparameters?
- Did you consider automated approaches to optimise the hyperparameters of the model architecture? If so, please provide details. If not, why were such methods not considered?
- On lines 859 – 873, the method of selecting the value of k used in the experiments is provided. The analysis was done on one specific dataset. However:
- It is unclear whether the same K can be used for other datasets, or another analysis has to be conducted. If an analysis needs to be conducted each time, is it a process the user has to do manually or is it automated?
- What is the motivation behind using the Silhouette Score for this assessment? Did you consider other scores such as Calinski-Harabasz index?
- Did you consider clustering algorithms that do not require explicitly setting K and allow more flexible clustering?

If my questions and the points in the “Weaknesses” section are addressed, I am happy to raise my score.

---

> ### Author Response · Authors · 2025-11-20
>
> We sincerely appreciate your constructive suggestions which will substantially improve our work.
>
> ---
>
> ### **Weakness 1**: Incorrect usage of “\citep” and “\citet” across the paper.
> ### **Weakness 2**: The appendix section is not properly numbered. The appendix should have letter “A, B, …” instead of numbers. Some acronyms are used without being presented or before being fully presented. Examples: NTAP, GCN.
>
> We have corrected the formatting issues in the revised version.
>
> ---
>
> ### **Weakness 3**: Very long introduction. The current introduction is very longs and includes both the introduction and the related work. It is better to have these sections separated for readability.
>
> Thank you for the suggestion. We have reorganized the introduction and separated the related work section (due to page limits this additional related work from the introduction is added to the Appendix B.1).
>
> ---
>
> ### **Weakness 3**: Unclear setting of hyperparameters. The authors do not provide details on how the hyperparameters of the model are set, nor the reasoning behind setting these hyperparameters. There is only a brief description in line 774-779.
> ### **Question 4**: Q4 - In lines 773–779, a brief description of setting the hyperparameters is presented. However, only a few hyperparameters are presented without the reasoning behind setting these values. Can you please elaborate on why setting these exact values and provide details and reasoning about other hyperparameters?
>
> Regarding the hyperparameter settings, we set the embedding size to 64. The rationale is: 1.our goal is to obtain a compact representation that offers computational speed-ups after dimensionality reduction while still preserving essential trajectory information, 2. ensure fair comparison with prior methods. For example, TrajCL uses an embedding dimension of 256, NEUTRAJ uses 128, and ST2Vec uses 64. Choosing 64 avoids giving our method an unfair advantage from a larger latent space and keeps the computational cost low and consistent across methods. For the batch size and learning rate, we adopted the values that produced the most stable training behavior on the T-Drive dataset(validation dataset). We have added more context in Appendix D.4.
>
> ---
>
> ### **Weakness 4**: The baseline methods are not properly presented. The authors compared their experiments with 4 baselines methods. However, the methods are not presented in details. It is important to present the details of these methods, regarding the architecture and the type of loss function they use. This assists in evaluating and understanding the experimental setup, in addition to clarifying the motivation of choosing these exact baselines.
>
> We have added the baseline details in Appendix D.3. To comprehensively evaluate the effectiveness of our method in trajectory representation, we compare it against four representative models: ConDTC, ST2Vec, NeuTraj, and TrajCL. These methods cover the major technical directions in current trajectory research:
>  - ConDTC: Combines a Transformer encoder with location and temporal embeddings, together with contrastive clustering, for fine-grained spatio-temporal pattern recognition.
>  - TrajCL: Utilizes multiple trajectory augmentation strategies, contrastive learning, and an improved dual-feature self-attention encoder, representing contrastive trajectory representation learning approaches.
>  - ST2Vec: Employs triplet loss and a spatio-temporal co-attention mechanism to emphasize temporal–spatial details in similarity computation, serving as a key method for time-sensitive similarity modeling.
>  - NeuTraj: Uses an RNN backbone with a distance-weighted ranking loss to preserve similarity order, and is often applied for approximating high-cost similarity measures such as Frechet and DTW.
>
> The four selected models cover different technical directions, including Transformer-based, self-attention–based, and RNN-based encoders, along with a variety of loss functions (contrastive, triplet, ranking, and distance-weighted ranking). Together, they form a representative set of baselines for trajectory analysis. We chose them because each reflects an important research direction, enabling a comprehensive and balanced comparison.
>
> Our evaluation spans contrastive and triplet losses, ranking losses, and distance-weighted ranking mechanisms, while the model architectures range from RNN/LSTM/GRU to Transformer and Dual-feature Self-attention encoders, including spatio-temporal attention modules designed specifically for trajectory data. This ensures that our assessment does not favor any single line of methodology. We also include models that rely on data augmentation, and the results show that our method achieves the best performance across all metrics even without using augmentation, demonstrating strong generalization and data efficiency. In summary, the experimental results and AB tests clearly indicate that our proposed encoder and loss design provide consistent advantages.

---

> ### Author Response · Authors · 2025-11-20
>
> ### **Question 1**: Are there specific threshold to control these selections? If so, can you please elaborate on this? Without thresholds, it can happen that the positive sample is too close or identical to the anchor, and/or the negative sample is too far from the anchor. This hinders the effectiveness of contrastive triplet losses.
>
> How the framework was designed given the two points mentioned below, (1) A Structure-Preserving Embedding Framework Without Collapse and (2) A Hyperparameter-Free Learning Strategy for Triplet Learning. The selection (triplet construction) is based on the motivation defined in (2). And as you mentioned the following “positive sample is too close or identical to the anchor, and/or the negative sample is too far from the anchor” as it’s a common collapse issue for contrastive/triplet learning we elaborate more in (1).
>
> ### **1. A Structure-Preserving Embedding Framework Without Collapse**
> We propose a framework composed of a training strategy and the DMT Loss that enables the embedding to successfully recover the original structure, as shown in Figure 4(c), effectively avoiding issues such as:
> - **Dimension collapse** [1] in contrastive learning (e.g., TrajCL in Figure 4(b)), where for example positive samples are pulled excessively close and negative samples are pushed extremely far away.
> - **Class collapse** [2] in triplet learning (e.g., ST2Vec in Figure 4(a)), where performance heavily depends on the margin setting and requires mining strategies or extensive effort to handle hard negative triplets. This often produces clusters, but samples within the same cluster become difficult to distinguish (poor intra-class separation).
>
> Thus, our framework removes the dimension collapse behavior of traditional contrastive learning and the class collapse behavior of triplet learning. By replacing relative ranking with distance-pair alignment, adopting multi-scale distance alignment, and using the batch median distance to eliminate outliers, the model through the DMT Loss preserves both local and global geometric relationships of trajectories and successfully maintains the original structure.
>
> ### **2. A Hyperparameter-Free Learning Strategy for Triplet Learning**
> Our framework completely eliminates the core hyperparameters required in contrastive learning or triplet learning. Through the design of the DMT Loss and our training strategy that uses top-1 positive and random negative sampling, the model automatically maintains negative-sample diversity, learns both local/global relationships, and preserves the original structure.
>
> Baselines listed in Table 1, as well as common contrastive/triplet approaches, typically depend on hyperparameters such as the number/proportion of positive and negative samples, margins/temperatures, mining strategies, and data-augmentation settings [3,4,5] —factors that often distort representations and lead to collapse.
>
> Our method removes margins, temperatures, mining strategies, and data augmentation entirely, and avoids manually specifying the positive/negative ratio. The triplet construction in our training strategy, grounded in the DMT Loss, is instead motivated by two key concepts:
> - **"A person is defined by the friend closest to them."** [6] : In the model, each anchor learns only from its most similar top-1 positive. If two samples mutually select each other as their top-1, their relationship is already sufficiently strong, and the model does not need to overly compress their distance.
> - **"You are defined by your closest friend, but you can understand the world through strangers."** [7] : Negatives are randomly chosen from samples that are neither the anchor nor its top-1 positive. This allows the model to learn global structure from a diverse set of dissimilar samples without ever specifying the number of negatives or relying on negative mining.
>
> Additionally, Table 3(b) presents stability experiments with smaller training sets. Even when the dataset is reduced by 50% which decreases triplet diversity, the performance drops only marginally, indicating strong robustness to sample diversity.
>
> [1] Jing et al. Understanding Dimensional Collapse in Contrastive Self-supervised Learning. https://arxiv.org/abs/2110.09348.
>
> [2] Xuan et al. Hard negative examples are hard, but useful. https://arxiv.org/abs/2007.12749.
>
> [3] He et al. Momentum Contrast for Unsupervised Visual Representation Learning. https://arxiv.org/abs/1911.05722.
>
> [4] Schroff et al. FaceNet: A Unified Embedding for Face Recognition and Clustering. https://arxiv.org/abs/1503.03832.
>
> [5] Chen et al. A Simple Framework for Contrastive Learning of Visual Representations. https://arxiv.org/abs/2002.05709.
>
> [6] Zhou et al. Discrete Hierarchical Organization of Social Group Sizes. https://arxiv.org/abs/cond-mat/0403299.
>
> [7] Leskovec and Horvitz. Planetary-Scale Views on an Instant-Messaging Network. https://arxiv.org/abs/0803.0939

---

> ### Author Response · Authors · 2025-11-20
>
> ### **Weakness 6**: The code is not available for the review process. The authors mention in the paper abstract that “An anonymous repo with code and data is in the supplement.”. However, there is no such link in the appendix.
>
> The badminton dataset is derived from video annotation and then converted into the BLSR sequence structure. The public dataset link is available at:https://github.com/wywyWang/CoachAI-Projects/tree/main/ShuttleSet.
>
> For the taxi trajectory datasets, the public links are:
> https://www.kaggle.com/datasets/arashnic/tdriver,
> https://ieee-dataport.org/open-access/crawdad-romataxi.
>
> We have already uploaded the full source code and all datasets used in our experiments as a compressed package in the Supplementary Material. An anonymous GitHub repository is also provided here:
> https://anonymous.4open.science/r/TRIDENT-D877
>
> After the process, we will publicly release the complete codebase and all experiment-ready datasets.
>
> ---
>
> ### **Question 2**: Is the computation cost considered? And are there are pre-computing steps to ensure that the same calculation between the exact same sample is not done many times? This is important for the scalability of the proposed method.
>
> Computation cost isn’t specifically considered, but if we focus on the formation of triplets and the calculation of the loss:
>  - **O(N² * M²) for triplets pre-construction**: We have a pre-computing step that computes the TP between all N data pairs, and for each anchor we select only the top-1 positive and a random negative. This results in roughly O(N² * M²) time complexity (where M is the average trajectory length, which can be ignored if it is small).
>
>  - **O(B * d) for loss when training**: For the DMT loss, since it directly uses the pre-computed triplets, there is no additional time spent during training to form triplets; only the loss itself needs to be computed, which is roughly O(B * d), where B is the batch size and d is the dimension size.
>
> Also, in the pre-computing step, because each anchor + top-1 positive pair is matched with only one random negative, each triplet is unique.
>
> ---
>
> ### **Question 3**:  In lines 250–257, the authors discuss the values of k . However, there is no discussion nor mathematical notation showing how it is calculated. Can you please provide more details?
>
> In the DMT loss, $\kappa$ defines the number of multipliers $m$. Since these are scaling factors relative to the batch median distance $σ_{base}$, we fix $m = [0.5, 1.0, 2.0]$, which means $\kappa = 3$. $0.5$ in $m$ emphasizes the local neighborhood and $2.0$ considers toward the global structure, and $1.0$ is somewhere in between.
>
> ---
>
> ### **Question 5**: Did you consider automated approaches to optimise the hyperparameters of the model architecture? If so, please provide details. If not, why were such methods not considered?
>
> We did consider automated hyperparameter optimization, but found it unnecessary. The framework already eliminates many commonly criticized hyperparameters in triplet learning (more details in Response to Reviewer D5b8 Question 1 “2. A Hyperparameter-Free Learning Strategy for Triplet Learning”).
>
> This leaves only four hyperparameters: batch size, learning rate, number of self-attention layers, and embedding size. The embedding size strongly affects trajectory similarity computation speed, so we follow common baseline practice and keep it below 64 (though overly small sizes may lose information). The remaining parameters are basic settings determined via the validation set. We have added two additional figures in Appendix D.4.
>
> ---
>
> ### **Question 6**: It is unclear whether the same K can be used for other datasets, or another analysis has to be conducted. If an analysis needs to be conducted each time, is it a process the user has to do manually or is it automated?
>
> The value of K cannot be reused across different datasets, because the movement patterns, sample distributions, and clustering structures different. Therefore, the most appropriate K must be determined separately for each. In our workflow, this step is automated. Evaluates a set of candidate K values, computes the Silhouette Score for each, and selects the one with the highest score. This selection of K is a common approach in trajectory clustering and general unsupervised learning.
>
> ---
>
> ### **Question 7**: What is the motivation behind using the Silhouette Score for this assessment? Did you consider other scores such as Calinski-Harabasz index?
>
> We use the Silhouette Score because our embedding preserves meaningful geometric distances, and Silhouette evaluates clustering quality based on sample-to-sample distances, which aligns with our representation. Other metrics (e.g., the Calinski–Harabasz index) are also valid but focus more on different aspects. Overall, our framework does not rely on any specific metric and can be adapted to others.

---

> ### Author Response · Authors · 2025-11-20
>
> ### **Question 8**: Did you consider clustering algorithms that do not require explicitly setting K and allow more flexible clustering?
>
> We have considered clustering methods that do not require specifying K and agree that offer flexibility. Our current use of K-based clustering follows the settings in existing trajectory-clustering literature (ConDTC[1] initialize the cluster center vectors C by applying K-means, K can be determined by the elbow method).
>
> However, we have also begun exploring how our framework can be extended to scenarios where the number of k does not need to be specified. Instead, our aim is to learn a more structured embedding so that density-based or structure-based clustering loss methods can naturally form groups without requiring a predefined K. This direction fits well with our framework, and we appreciate the reviewer’s suggestion as it aligns with our future work.
>
> [1] J. Si et al. ConDTC: Contrastive Deep Trajectory Clustering for Fine-Grained Mobility Pattern Mining. https://ieeexplore.ieee.org/document/10433249

---

> > ### Author Response · Authors · 2025-12-02
> >
> > ### **Weakness 5**: When comparing the framework presented in the paper with the baselines, the results are presented in numerical forms without testing whether the improvements are statistically significant or not. The statistical significance of the claimed improvements in comparison with the baselines methods needs to be tested (using a Friedman test followed by Nemenyi post-hoc test, for example).
> >
> > We have added the required **Friedman test followed by Nemenyi post-hoc test** in Appendix D.8.
> >
> > Table 6 shows that our model consistently achieves rank 1 across all metrics and datasets compared to various baselines. To assess the statistical significance of these improvements, we first conducted the Friedman test to determine whether there are significant differences among the evaluated frameworks. The test yields a p-value of $1.078 \times 10^{-5}$, which is well below the conventional threshold of 0.05, indicating the presence of statistically significant differences. Consequently, we further performed a Nemenyi post-hoc test to identify which frameworks differ significantly from our method shown in Table 7.

---

> > > ### Author Response · Authors · 2025-12-02
> > >
> > > Dear Reviewer d5b8, Thank you very much for your positive evaluation during the initial review. We have now thoroughly addressed and resolved all the issues raised earlier. We understand that due to the recent OpenReview incident on anonymity, we have not been able to receive additional affirmation and suggestions from you.

---

### Official Review · Reviewer_LWGz · 2025-10-31

**Soundness:** 3
**Presentation:** 3
**Contribution:** 3
**Rating:** 4
**Confidence:** 5

**Summary:**

This paper presents TRIDENT, a unified framework for learning spatio-temporal trajectory representations. It handles diverse data types—from smooth GPS routes to abrupt sports movements—within a single model. TRIDENT integrates: (1) a Graph Convolutional Network (GCN) for spatial features and a temporal embedding module fused via a Dual-Attention Encoder (DAEncoder); (2) a Distance-preserving Multi-kernel Triplet Loss (DMT) that maintains the metric geometry of the original trajectory space; and (3) extensive experiments on T-Drive, Rome, and Badminton datasets showing superior retrieval accuracy, efficiency, and cross-domain generalization. A case study on badminton rallies further demonstrates the model’s ability to reveal tactical patterns.

**Strengths:**

S1: The proposed Distance-preserving Multi-kernel Triplet Loss (DMT) addresses a fundamental limitation of classic triplet loss by enforcing the preservation of metric structure, not just relative ordering.

S2: TRIDENT effectively handles diverse trajectory types—from smooth taxi routes to abrupt sports movements—within a unified model.

**Weaknesses:**

W1: The overall architecture (“GCN + temporal model + attention fusion”) follows a well-established design pattern in spatiotemporal modeling, which somewhat limits the novelty of the framework.

W2: The project's code and data are not publicly available, and the datasets used for training and testing the models are too small.

W3: The paper lacks comparisons with general sequence representation learning models, which weakens its state-of-the-art (SOTA) conclusions.

**Questions:**

Q1: Could you please clarify what "D2V" (line 129) refers to in the context of your temporal embedding module?

Q2: What is the meaning of the “LEARNING PROCESS OF” in section3.1?

---

> ### Author Response · Authors · 2025-11-20
>
> We appreciate your valuable comments and would like to address each of your concerns.
>
> ---
>
> ### **Weakness 1**: The overall architecture (“GCN + temporal model + attention fusion”) follows a well-established design pattern in spatiotemporal modeling, which somewhat limits the novelty of the framework.
>
> We agree that trajectory representation methods generally rely on spatial and temporal features or some fusion and typically adopt contrastive or triplet learning, even in the four baselines. In line with this, our approach places particular emphasis on triplet construction, the loss function, the learned embedding geometry structure, and the treatment of complex trajectories. Existing methods in Table 1 exhibit several limitations: (1) the learned embeddings may fail to preserve the underlying structure (i.e., they may collapse), (2) contrastive/triplet learning is often highly sensitive to hyperparameters, and (3) prior methods perform reasonably well on continuous trajectories but are less effective for discrete trajectories.
>
> Here, we present three novel contributions in our proposed approach, each addressing one of the issues above. More details for each limitations and the corresponding solution are described below:
> ### **1. A Structure-Preserving Embedding Framework Without Collapse**
> We propose a framework composed of a training strategy and the DMT Loss that enables the embedding to successfully recover the original structure, as shown in Figure 4(c), effectively avoiding issues such as:
> - **Dimension collapse** [1] in contrastive learning (e.g., TrajCL in Figure 4(b)), where for example positive samples are pulled excessively close and negative samples are pushed extremely far away.
> - **Class collapse** [2] in triplet learning (e.g., ST2Vec in Figure 4(a)), where performance heavily depends on the margin setting and requires mining strategies or extensive effort to handle hard negative triplets. This often produces clusters, but samples within the same cluster become difficult to distinguish (poor intra-class separation).
>
> Thus, our framework removes the dimension collapse behavior of traditional contrastive learning and the class collapse behavior of triplet learning. By replacing relative ranking with distance-pair alignment, adopting multi-scale distance alignment, and using the batch median distance to eliminate outliers, the model through the DMT Loss preserves both local and global geometric relationships of trajectories and successfully maintains the original structure.

---

> ### Author Response · Authors · 2025-11-20
>
> ### **2. A Hyperparameter-Free Learning Strategy for Triplet Learning**
> Our framework completely eliminates the core hyperparameters required in contrastive learning or triplet learning. Through the design of the DMT Loss and our training strategy that uses top-1 positive and random negative sampling, the model automatically maintains negative-sample diversity, learns both local/global relationships, and preserves the original structure.
>
> Baselines listed in Table 1, as well as common contrastive/triplet approaches, typically depend on hyperparameters such as the number/proportion of positive and negative samples, margins/temperatures, mining strategies, and data-augmentation settings [3,4,5] —factors that often distort representations and lead to collapse.
>
> Our method removes margins, temperatures, mining strategies, and data augmentation entirely, and avoids manually specifying the positive/negative ratio. The triplet construction in our training strategy, grounded in the DMT Loss, is instead motivated by two key concepts:
> - **"A person is defined by the friend closest to them."** [6] : In the model, each anchor learns only from its most similar top-1 positive. If two samples mutually select each other as their top-1, their relationship is already sufficiently strong, and the model does not need to overly compress their distance.
> - **"You are defined by your closest friend, but you can understand the world through strangers."** [7] : Negatives are randomly chosen from samples that are neither the anchor nor its top-1 positive. This allows the model to learn global structure from a diverse set of dissimilar samples without ever specifying the number of negatives or relying on negative mining.
>
> Additionally, Table 3(b) presents stability experiments with smaller training sets. Even when the dataset is reduced by 50% which decreases triplet diversity, the performance drops only marginally, indicating strong robustness to sample diversity.

---

> ### Author Response · Authors · 2025-11-20
>
> ### **3. A Generalizable Attention Pooling Mechanism for Both Continuous and Discrete Trajectories**
> We introduce Nonlinear Tanh-Projection Attention Pooling (NTAP), a mechanism capable of handling both continuous and discrete trajectories (well-defined under Response to Reviewer 719K Weakness 2/Question 2), addressing the limitations of standard attention pooling (Std-AP) in discrete trajectory settings.
>
> As shown in Table 2(b), NTAP and Std-AP perform as follows:
> - **Continuous trajectories (T-Drive Dataset)**: NTAP performs on par with Std-AP, indicating that NTAP successfully preserves major motion patterns in continuous trajectories.
> - **Discrete trajectories (Badminton Dataset)**: NTAP outperforms Std-AP, demonstrating its superior ability to capture the characteristics of discrete movements.
>
> Traditional attention pooling uses linear projection or raw features to compute attention scores. This works well for continuous trajectories with continuous, no-reset, and constantly adjusted behavior, but causes problems for discrete trajectories with event-triggered movement, return points, and episodic structures as described below:
> - **Lack of higher-order interactions (weak nonlinear modeling)**: For continuous trajectories that rely heavily on nonlinear changes in speed, acceleration, or direction, linear attention cannot model these relationships effectively. For discrete movements with sharp zig-zags or high-curvature turns, attention scores easily average out across time steps, diluting critical events.
> - **Insensitivity to episodic/fragmented events**: Return points or action-triggering events in discrete trajectories may be treated as ordinary time steps, preventing attention from emphasizing them appropriately.
>
> NTAP addresses both issues while still capturing the primary patterns of continuous movement. Its attention-score mechanism includes:
> - **Nonlinear projection ($\tanh\bigl(\boldsymbol{W}_\omega^\top \boldsymbol{z}_i\bigr)$)**: Maps each feature into a higher-order nonlinear space. For continuous trajectories, it preserves smooth variations and curved motion. For discrete trajectories, it highlights sharp turns and high-curvature event points.
> - **Learnable context vector($\boldsymbol{u}_\omega$)**: Models nonlinear interactions across dimensions, automatically identifying important time steps or positions. Event-triggered points and return points in discrete movements are more easily emphasized.
>
> [1] Jing et al. Understanding Dimensional Collapse in Contrastive Self-supervised Learning. https://arxiv.org/abs/2110.09348.
>
> [2] Xuan et al. Hard negative examples are hard, but useful. https://arxiv.org/abs/2007.12749.
>
> [3] He et al. Momentum Contrast for Unsupervised Visual Representation Learning. https://arxiv.org/abs/1911.05722.
>
> [4] Schroff et al. FaceNet: A Unified Embedding for Face Recognition and Clustering. https://arxiv.org/abs/1503.03832.
>
> [5] Chen et al. A Simple Framework for Contrastive Learning of Visual Representations. https://arxiv.org/abs/2002.05709.
>
> [6] Zhou et al. Discrete Hierarchical Organization of Social Group Sizes. https://arxiv.org/abs/cond-mat/0403299.
>
> [7] Leskovec and Horvitz. Planetary-Scale Views on an Instant-Messaging Network. https://arxiv.org/abs/0803.0939

---

> ### Author Response · Authors · 2025-11-20
>
> ### **Weakness 2**: The project's code and data are not publicly available, and the datasets used for training and testing the models are too small.
>
> The badminton dataset is derived from video annotation and then converted into the BLSR sequence structure. The public dataset link is available at:https://github.com/wywyWang/CoachAI-Projects/tree/main/ShuttleSet.
>
> For the taxi trajectory datasets, the public links are:
> https://www.kaggle.com/datasets/arashnic/tdriver,
> https://ieee-dataport.org/open-access/crawdad-romataxi.
>
> We have already uploaded the full source code and all datasets used in our experiments as a compressed package in the Supplementary Material. An anonymous GitHub repository is also provided here:
> https://anonymous.4open.science/r/TRIDENT-D877
>
> After the process, we will publicly release the complete codebase and all experiment-ready datasets.
>
> ---
>
> ### **Weakness 3**: The paper lacks comparisons with general sequence representation learning models, which weakens its state-of-the-art (SOTA) conclusions.
>
> In Section 4.3.2 (AB Test), we compared several common and widely used sequence models (Transformer, BiLSTM, GRU) under the same GCN and temporal encoding settings. The results consistently show that our proposed DAE outperforms these sequence baselines, indicating that sequence modeling is not the bottleneck for this task and that our architecture better captures the spatiotemporal dependencies.
>
> Prior studies have also highlighted the limitations of general sequence models. For example, ST2Vec explicitly states that state-of-the-art RNN/LSTM models fail to capture periodic and non-periodic temporal patterns, and therefore uses a modified LSTM. TrajCL adopts an improved Dual-feature Self-attention Encoder(Transformer-based), and their experiments show that a vanilla Transformer cannot outperform TrajCL. NeuTraj uses an RNN-based backbone.
>
> These findings indicate that our comparisons against baseline already cover the representative families of general sequence representation learning models in trajectory research (RNN/LSTM/BiLSTM and Transformer). The mainstream architectures have already been included and shown to underperform our method. Moreover, we also compared the two dominant contrastive and triplet-based frameworks. Further justification of our baseline choices can be found in our response to Reviewer d5b8 Weakness 4.
>
> ---
>
> ### **Question 1**: Could you please clarify what "D2V" (line 129) refers to in the context of your temporal embedding module?
>
> “D2V” is Date2Vec. The revision has been made.
>
> ---
>
> ### **Question 2**: What is the meaning of the “LEARNING PROCESS OF” in section3.1?
>
> This issue was due to a formatting problem, which has now been corrected in the paper.

---

> > ### Comment · Reviewer_LWGz · 2025-11-27
> > **Response to Authors's Rebuttal**
> >
> > Thank you for the rebuttal. However, two concerns remain. First, the motivation for combining badminton trajectories with continuous GPS data is still not clearly justified; the two domains differ substantially, and the practical need for a unified representation is unclear. Second, the datasets used are relatively small, which limits the strength of the robustness claims. Evaluating on a larger dataset (e.g., the Chengdu dataset used in TrajCL) would provide more convincing evidence. Therefore, I tend to maintain the score.

---

> ### Author Response · Authors · 2025-12-02
>
> Dear Reviewer LWGz, thank you for the suggestions, we have addressed your concerns.
>
>
> **Your Second Concern**
>
> To address your concern on the dataset, we additionally further validate the robustness and generalizability of our model on even larger datasets, Chengdu and Xian in Appendix D.7. The performance improves over SOTA baselines via 21.1% in Chengdu and 24.1% in Xian are greater than those observed in the original data setting (11.2% in T-Drive and 10.9% in Rome). These results not only confirm the model’s stability under data scaling but also demonstrate its strong ability to generalize across diverse domains. We include all five datasets used in this study in the updated Supplementary Material (OpenReview).
>
> Link for Chengdu and Xian: https://www.kaggle.com/datasets/ash1971/didi-dataset?resource=download or https://outreach.didichuxing.com/appEn-vue/dataList
>
> **Your First Concern**
>
> Based on our discussion and the latest suggestions from Reviewer 719K, we now have a clearer and more formal mathematical definition of the spectrum of trajectory properties (see Trajectory Characteristics in Section 2 Preliminaries, Appendix D.2, and the discussion with Reviewer 719K). It is exactly why the two extremes of Urban Mobility and Badminton differ substantially, making an unified representation worthwhile. The reasons follow below.
>
> **Motivation**:
> - Deep Learning: An unified higher generalizability architecture (especially representations) can serve as a consistent backbone or be extended further to handle additional unknown trajectory patterns. In contrast, separate architectures tend to overfit to individual domains and lack scalability when encountering new trajectory types.
> - Scientific: Unifying the architecture minimizes system-level fragmentation and enables methodological consistency.
> - Engineering: Unified systems reduce engineering and maintenance burden. By comparison, additional duplicated development environments, increased cost and more error sensitivity for separate systems.
>
> **Practical Need**:
> The unified architecture provides better future extensibility, offers higher generality and robustness. With the same architecture on different trajectory datasets provides consistent representations, this can be used in clustering, analysis and even as embeddings for other models. We have provided additional experiments on Chengdu to address your concern, and even take a step further on Xian which proves further the practical need for such unified architecture that can easily extend to different trajectory domains. Additionally, a single model family eliminates the divergence of training pipelines, parameterization schemes, and optimization heuristics, which in turn reduces confounding factors in empirical studies. This becomes more interpretable and scientifically comparable. Also, in real-world company production deployment, teams avoid duplicated pipelines, fragmented codebases, and inconsistent update paths. This not only lowers development and operational cost but also ensures that bug fixes, feature improvements, and infrastructure upgrades automatically benefit all domains.
>
> Without a unified architecture, everyone ends up reinventing the wheel, producing results that only work in their own isolated environment. Which without shared architectures and standards would stagnate just like an overfitted model. Unification isn’t optional; it is the foundation for results that are usable, comparable, and scalable.

---

### Official Review · Reviewer_M4PH · 2025-10-31

**Soundness:** 3
**Presentation:** 3
**Contribution:** 3
**Rating:** 6
**Confidence:** 3

**Summary:**

This paper presents TRIDENT, a spatio-temporal trajectory representation learning framework based on a novel variant of contrastive learning. The approach is designed to be generalizable across different trajectory domains, demonstrated here on GPS trajectories and badminton (human movement) trajectories; the former being smooth and dense, and the latter sparse, abrupt, and event-driven.

To achieve this, the authors propose: (1) a GCN for spatial encoding and D2V for temporal embedding, (2) a Dual-Attention Encoder with tanh attention pooling to fuse spatio-temporal representations, and (3) a Distance-preserving Multi-kernel Triplet Loss (DMT). The DMT component is carefully designed to (a) preserve the geometric structure of the original trajectories and (b) leverage an anchor-positive-negative triplet loss setup, which is known to enhance the typical contrastive positive-negative pairs.

Experimental results show that TRIDENT achieves substantial improvements over baseline methods across three datasets spanning two domains. Qualitatively, TRIDENT trajectory representations are distributed in a way that facilitates easier clustering.

**Strengths:**

The authors carefully designed the model architecture and training strategy, tailored to capture the unique characteristics and complexities of spatio-temporal trajectories and their diverse domains due to varying collection methods and details.

The proposed DMT loss is an elegant adaptation of the triplet loss framework, which has proven effective in other domains such as text embeddings.

The model demonstrates strong cross-domain generalizability and achieves state-of-the-art performance compared to existing baselines.

The faster training/convergence is a nice benefit.

**Weaknesses:**

- The paper lacks cross-domain or cross-dataset transfer experiments. It would have been valuable to see pretraining on one domain (e.g., GPS) followed by evaluation or fine-tuning on another (e.g., badminton), OR training on one dataset (e.g., Beijing) and testing on another within the same domain (e.g., Rome).

- The evaluation focuses primarily on trajectory retrieval. Incorporating additional evaluation tasks, e.g., robustness tests under trajectory downsampling or slight distortions, would strengthen the overall empirical validation.

- Minor presentation issues:
Citation formatting is occasionally incorrect; for example, missing brackets around in-line citations makes them harder to read.
L55-66 “raw trajectories are embedding space” should be revised to “raw trajectories are encoded into the embedding space.”?
The title of Section 4.3.1 appears to contain a typographical error: “Learning Process of”…?

**Questions:**

How was the badminton trajectory dataset collected? Providing more details on the data collection process would improve clarity and reproducibility.

How does the proposed method perform under cross-domain or cross-dataset transfer settings?

Does the method remain robust when trajectories are distorted or downsampled? For example, an analysis simulating GPS errors (such as those caused by urban canyons, as the authors mentioned in the introduction) would also help evaluate TRIDENT’s robustness.

**Details Of Ethics Concerns:**

No ethical concerns noticed.

---

> ### Author Response · Authors · 2025-11-20
>
> Thank you for your insightful comments. Your feedback is extremely helpful, and we are committed to addressing each question you have raised.
>
> ---
>
> ### **Weakness 1**: The paper lacks cross-domain or cross-dataset transfer experiments. It would have been valuable to see pretraining on one domain (e.g., GPS) followed by evaluation or fine-tuning on another (e.g., badminton), OR training on one dataset (e.g., Beijing) and testing on another within the same domain (e.g., Rome).
> ### **Question 2**: How does the proposed method perform under cross-domain or cross-dataset transfer settings?
>
> We clarify the distinctions among cross-domain, cross-dataset, and cross-task first as they apply to our datasets:
> - **Cross-domain**: the task is the same, but the training and test distributions differ (e.g., due to road topology, traffic patterns, or mobility behavior).
> - **Cross-dataset**: data come from different sources or cities; in trajectory problems, this typically coincides with cross-domain shifts.
> - **Cross-task**: the tasks, label spaces, and trajectory semantics differ fundamentally.
>
> For example, T-Drive and Rome represent the same task (urban taxi mobility) but originate from different cities with different traffic regimes and road networks. Therefore, this setting is cross-dataset and cross-domain, but not cross-task. In contrast, Badminton trajectories differ from taxi trajectories in scale, movement dynamics, semantic meaning, and downstream objectives. Thus, Badminton vs. T-Drive/Rome constitutes a cross-task setting, not merely cross-domain.
>
> Our method exhibits strong domain-general capability: all experiments use the same architecture and identical hyperparameters, and the model consistently handles three fundamentally different datasets (Badminton, T-Drive, and Rome). To further address your concern, we conducted additional cross-domain transfer experiments: (1) training on T-Drive and testing on Rome, and vice versa. Baselines were included for comparison.
>
> Across all settings, our method outperforms the baselines. Although performance drops under cross-dataset or cross-domain transfer, the decline is well explained by inherent data differences rather than limitations of the model. First, even without GCNs, the geographic coordinate distributions of different cities are fundamentally distinct, and road topology, traffic rules, and movement constraints (e.g., one-way streets, intersection layout) vary substantially, making the trajectory distributions between cities non-alignable. Second, driving behaviors and common route structures differ by city, meaning that the same type of driving trajectory does not carry consistent semantics across domains. Third, GPS and badminton trajectories follow completely different motion mechanisms (Also discussed in our response to Reviewer 719K Weakness 2/Question 2).
>
> Despite these inherent gaps, our method indicates that it learns stable cross-dataset spatio-temporal representations. Overall, these additional experiments show that our framework remains competitive and that performance drops arise from fundamental differences in real-world trajectory distributions. The additional results have been added to Section 4.3.3.
>
> ---
>
> ### **Weakness 2**: The evaluation focuses primarily on trajectory retrieval. Incorporating additional evaluation tasks, e.g., robustness tests under trajectory downsampling or slight distortions, would strengthen the overall empirical validation.
> ### **Question 3**: Does the method remain robust when trajectories are distorted or downsampled? For example, an analysis simulating GPS errors (such as those caused by urban canyons, as the authors mentioned in the introduction) would also help evaluate TRIDENT’s robustness.
>
> We hope to provide more complete distorted or downsampled analysis results by December 3rd. Additionally, we have added stability experiments with multiple reduced size training sets in Section 4.3.3.
>
> ---
>
> ### **Weakness 3**: Minor presentation issues: Citation formatting is occasionally incorrect; for example, missing brackets around in-line citations makes them harder to read. L55-66 “raw trajectories are embedding space” should be revised to “raw trajectories are encoded into the embedding space.”? The title of Section 4.3.1 appears to contain a typographical error: “Learning Process of”…?
>
> We have corrected the formatting issues in the revised version.

---

> ### Author Response · Authors · 2025-11-20
>
> ### **Question 1**: How was the badminton trajectory dataset collected? Providing more details on the data collection process would improve clarity and reproducibility.
>
> The dataset was collected from badminton match videos. The specific collection process involves performing annotation on the videos, and ultimately storing it in a sequential CSV file format [1,2]. These CSV files can be obtained from the public data link :https://github.com/wywyWang/CoachAI-Projects/tree/main/ShuttleSet.
> , the Supplementary Material link uploaded on OpenReview, the code and dataset compressed file provided by us, and the anonymous GitHub link (https://anonymous.4open.science/r/TRIDENT-D877).
>
> [1] Wang et al. ShuttleSet: A Human-Annotated Stroke-Level Singles Dataset for Badminton Tactical Analysis. https://arxiv.org/abs/2306.04948.
>
> [2] Wang et al. Benchmarking stroke forecasting with stroke-level badminton dataset. https://www.ijcai.org/proceedings/2024/1042.pdf.

---

> ### Author Response · Authors · 2025-12-02
>
> Dear Reviewer M4PH, thank you for the suggestions, we have respond to all your questions.

---

### Official Review · Reviewer_719K · 2025-11-05

**Soundness:** 2
**Presentation:** 2
**Contribution:** 2
**Rating:** 2
**Confidence:** 4

**Summary:**

This paper proposes a unified trajectory representation learning framework designed to handle both continuous movement trajectories and discrete trajectories. Experiments on several datasets demonstrate the effectiveness of the proposed method for downstream tasks such as trajectory similarity search.

**Strengths:**

* The paper tackles an interesting problem of learning representations that generalize across both continuous and discrete trajectory types.

* The experimental results show that the proposed framework performs well on specific datasets, indicating potential for improving trajectory similarity search.

**Weaknesses:**

1. At first, I thought the proposed framework aimed to learn a unified trajectory representation by jointly learning from all types of trajectory data (e.g., both badminton and taxi datasets). However, it appears that the model is still trained separately for each data type. In that case, I do not see a clear advantage of this approach compared to existing methods. The overall model design (e.g., handling spatial and temporal embeddings separately and then combining them), together with the use of a triplet loss, is already widely adopted in trajectory representation learning.

2. The paper should clearly define the fundamental difference between continuous and discrete movements. Is the distinction based on predictability or on sampling rate? If the latter, then directly resampling the taxi trajectories could produce a similar dataset, suggesting that the distinction may not be substantial. A formal definition and discussion are necessary to clarify this point and justify the motivation.

3. The writing quality and organization need improvement. The introduction is overly long and occupies too much space, while the method section is too brief and lacks sufficient implementation details, making it difficult to reproduce the work. Additionally, the topic may be too narrow in scope and might not appeal to the broader ICLR readers.

**Questions:**

1. What is the novelty of the proposed approach, given that most components of the model are already well studied in the literature?

2. What is the fundamental difference between continuous and discrete movements? Why do they need to be treated separately, and which parts of the model explicitly account for this distinction? A clearer conceptual and architectural explanation would strengthen the contribution.

3. Since discrete trajectories are the main focus of the paper and show the most performance improvement, the evaluation should include more relevant datasets, such as basketball or soccer trajectories, to better validate the generality of the proposed approach.

---

> ### Author Response · Authors · 2025-11-20
>
> We thank the reviewer for their valuable feedback and provide responses below.
>
> ---
>
> ### **Weakness 1**: At first, I thought the proposed framework aimed to learn a unified trajectory representation by jointly learning from all types of trajectory data (e.g., both badminton and taxi datasets). However, it appears that the model is still trained separately for each data type. In that case, I do not see a clear advantage of this approach compared to existing methods. The overall model design (e.g., handling spatial and temporal embeddings separately and then combining them), together with the use of a triplet loss, is already widely adopted in trajectory representation learning.
> ### **Question 1**: What is the novelty of the proposed approach, given that most components of the model are already well studied in the literature?
>
> We agree that trajectory representation methods generally rely on spatial and temporal features or some fusion and typically adopt contrastive or triplet learning, even in the four baselines. In line with this, our approach places particular emphasis on triplet construction, the loss function, the learned embedding geometry structure, and the treatment of complex trajectories. Existing methods in Table 1 exhibit several limitations: (1) the learned embeddings may fail to preserve the underlying structure (i.e., they may collapse), (2) contrastive/triplet learning is often highly sensitive to hyperparameters, and (3) prior methods perform reasonably well on continuous trajectories but are less effective for discrete trajectories.
>
> Here, we present three novel contributions in our proposed approach, each addressing one of the issues above. More details for each limitations and the corresponding solution are described below:
> ### **1. A Structure-Preserving Embedding Framework Without Collapse**
> We propose a framework composed of a training strategy and the DMT Loss that enables the embedding to successfully recover the original structure, as shown in Figure 4(c), effectively avoiding issues such as:
> - **Dimension collapse** [1] in contrastive learning (e.g., TrajCL in Figure 4(b)), where for example positive samples are pulled excessively close and negative samples are pushed extremely far away.
> - **Class collapse** [2] in triplet learning (e.g., ST2Vec in Figure 4(a)), where performance heavily depends on the margin setting and requires mining strategies or extensive effort to handle hard negative triplets. This often produces clusters, but samples within the same cluster become difficult to distinguish (poor intra-class separation).
>
> Thus, our framework removes the dimension collapse behavior of traditional contrastive learning and the class collapse behavior of triplet learning. By replacing relative ranking with distance-pair alignment, adopting multi-scale distance alignment, and using the batch median distance to eliminate outliers, the model through the DMT Loss preserves both local and global geometric relationships of trajectories and successfully maintains the original structure.

---

> ### Author Response · Authors · 2025-11-20
>
> ### **2. A Hyperparameter-Free Learning Strategy for Triplet Learning**
> Our framework completely eliminates the core hyperparameters required in contrastive learning or triplet learning. Through the design of the DMT Loss and our training strategy that uses top-1 positive and random negative sampling, the model automatically maintains negative-sample diversity, learns both local/global relationships, and preserves the original structure.
>
> Baselines listed in Table 1, as well as common contrastive/triplet approaches, typically depend on hyperparameters such as the number/proportion of positive and negative samples, margins/temperatures, mining strategies, and data-augmentation settings [3,4,5] —factors that often distort representations and lead to collapse.
>
> Our method removes margins, temperatures, mining strategies, and data augmentation entirely, and avoids manually specifying the positive/negative ratio. The triplet construction in our training strategy, grounded in the DMT Loss, is instead motivated by two key concepts:
> - **"A person is defined by the friend closest to them."** [6] : In the model, each anchor learns only from its most similar top-1 positive. If two samples mutually select each other as their top-1, their relationship is already sufficiently strong, and the model does not need to overly compress their distance.
> - **"You are defined by your closest friend, but you can understand the world through strangers."** [7] : Negatives are randomly chosen from samples that are neither the anchor nor its top-1 positive. This allows the model to learn global structure from a diverse set of dissimilar samples without ever specifying the number of negatives or relying on negative mining.
>
> Additionally, Table 3(b) presents stability experiments with smaller training sets. Even when the dataset is reduced by 50% which decreases triplet diversity, the performance drops only marginally, indicating strong robustness to sample diversity.

---

> ### Author Response · Authors · 2025-11-20
>
> ### **3. A Generalizable Attention Pooling Mechanism for Both Continuous and Discrete Trajectories**
> We introduce Nonlinear Tanh-Projection Attention Pooling (NTAP), a mechanism capable of handling both continuous and discrete trajectories (well-defined under Response to Reviewer 719K Weakness 2/Question 2), addressing the limitations of standard attention pooling (Std-AP) in discrete trajectory settings.
>
> As shown in Table 2(b), NTAP and Std-AP perform as follows:
> - **Continuous trajectories (T-Drive Dataset)**: NTAP performs on par with Std-AP, indicating that NTAP successfully preserves major motion patterns in continuous trajectories.
> - **Discrete trajectories (Badminton Dataset)**: NTAP outperforms Std-AP, demonstrating its superior ability to capture the characteristics of discrete movements.
>
> Traditional attention pooling uses linear projection or raw features to compute attention scores. This works well for continuous trajectories with continuous, no-reset, and constantly adjusted behavior, but causes problems for discrete trajectories with event-triggered movement, return points, and episodic structures as described below:
> - **Lack of higher-order interactions (weak nonlinear modeling)**: For continuous trajectories that rely heavily on nonlinear changes in speed, acceleration, or direction, linear attention cannot model these relationships effectively. For discrete movements with sharp zig-zags or high-curvature turns, attention scores easily average out across time steps, diluting critical events.
> - **Insensitivity to episodic/fragmented events**: Return points or action-triggering events in discrete trajectories may be treated as ordinary time steps, preventing attention from emphasizing them appropriately.
>
> NTAP addresses both issues while still capturing the primary patterns of continuous movement. Its attention-score mechanism includes:
> - **Nonlinear projection ($\tanh\bigl(\boldsymbol{W}_\omega^\top \boldsymbol{z}_i\bigr)$)**: Maps each feature into a higher-order nonlinear space. For continuous trajectories, it preserves smooth variations and curved motion. For discrete trajectories, it highlights sharp turns and high-curvature event points.
> - **Learnable context vector($\boldsymbol{u}_\omega$)**: Models nonlinear interactions across dimensions, automatically identifying important time steps or positions. Event-triggered points and return points in discrete movements are more easily emphasized.
>
> [1] Jing et al. Understanding Dimensional Collapse in Contrastive Self-supervised Learning. https://arxiv.org/abs/2110.09348.
>
> [2] Xuan et al. Hard negative examples are hard, but useful. https://arxiv.org/abs/2007.12749.
>
> [3] He et al. Momentum Contrast for Unsupervised Visual Representation Learning. https://arxiv.org/abs/1911.05722.
>
> [4] Schroff et al. FaceNet: A Unified Embedding for Face Recognition and Clustering. https://arxiv.org/abs/1503.03832.
>
> [5] Chen et al. A Simple Framework for Contrastive Learning of Visual Representations. https://arxiv.org/abs/2002.05709.
>
> [6] Zhou et al. Discrete Hierarchical Organization of Social Group Sizes. https://arxiv.org/abs/cond-mat/0403299.
>
> [7] Leskovec and Horvitz. Planetary-Scale Views on an Instant-Messaging Network. https://arxiv.org/abs/0803.0939

---

> ### Author Response · Authors · 2025-11-20
>
> ### **Weakness 2**: The paper should clearly define the fundamental difference between continuous and discrete movements. Is the distinction based on predictability or on sampling rate? If the latter, then directly resampling the taxi trajectories could produce a similar dataset, suggesting that the distinction may not be substantial. A formal definition and discussion are necessary to clarify this point and justify the motivation.
> ### **Question 2**: What is the fundamental difference between continuous and discrete movements? Why do they need to be treated separately, and which parts of the model explicitly account for this distinction? A clearer conceptual and architectural explanation would strengthen the contribution.
>
> The distinction between continuous and discrete movements in this work is not based on predictability or sampling rate, but follows established theories in the motor control literature. The fundamental difference lies in their temporal structure, control mechanisms, and whether there is a reset point during the behavior. In classic motor control research, Ghez et al. [8] showed that discrete movements have clear start and end points and are governed by event timing; Janzen et al. [9] further noted that discrete movements often return to a stable baseline posture after completion, giving them an episodic structure (Same study from Badminton dataset paper Wang et al. [10]). Movements in badminton match this pattern: each stroke is an independent event that triggers movement toward the shuttle’s landing point, and players usually return to the center location afterward, forming a repeated cycle. This characteristic also appears in the badminton heatmap figure(Wang et al. [11]), where the court center shows a clear concentration. Discrete movements can be defined by the following three characteristics:
> - **Event-triggered movement**: Movements are initiated and terminated by clear events, forming a segmented structure (Each movement is triggered by the opponent’s hitting action).
> - **Presence of a reset point**: After completing a movement, the player typically returns to a baseline position, making each segment independent. (This is also visible in badminton heatmaps)
> - **Episodic temporal organization**:  Each movement segment is completed as an independent unit, without continuous adjustment of targets or directions. (The next target point is usually approached directly, restarting the movement cycle.)
>
> In contrast, the essence of continuous movements is that the behavior does not naturally break into segments, there is no reset point between movements. Instead, the trajectory is continuously adjusted according to external conditions. Soccer and taxi trajectories are good examples. In soccer, players do not return to a fixed location after each offensive or defensive transition; rather, they continuously adjust their position based on teammates, opponents, spatial configuration, and tactical needs. Taxi trajectories follow the same pattern: drivers do not return to a fixed starting point after each ride, and their positions are dynamically updated by traffic, road networks, passenger demand, and dispatch information. In other words, both soccer and taxi movements show continuous, no-reset, and constantly adjusted behavior, making them typical continuous movements.
>
> Therefore, the difference between continuous and discrete movements comes from their behavioral nature and control structure, not from sampling rate. Changing the sampling density does not alter the inherent nature of these trajectories. We use this distinction as the foundation for cross-domain trajectory comparison. Moreover, even with any form of downsampling, upsampling, or resampling, taxi data will not show the sharp zig-zag patterns seen in badminton. The sampling rate only changes point density. It cannot generate high-curvature or large-angle directional changes (limited by road).
>
> [8]Ghez et al. Discrete and continuous planning of hand movements and isometric force trajectories.https://link.springer.com/article/10.1007/pl00005692#citeas
>
> [9]Janzen et al. Timing skills and expertise: discrete and continuous timed movements among musicians and athletes. https://www.researchgate.net/publication/260230278_Timing_skills_and_expertise_discrete_and_continuous_timed_movements_among_musicians_and_athletes
>
> [10] Wang et al. ShuttleSet: A Human-Annotated Stroke-Level Singles Dataset for Badminton Tactical Analysis. https://arxiv.org/abs/2306.04948.
>
> [11] Wang et al. Benchmarking stroke forecasting with stroke-level badminton dataset. https://www.ijcai.org/proceedings/2024/1042.pdf.

---

> ### Author Response · Authors · 2025-11-20
>
> ### **Weakness 3**: The writing quality and organization need improvement. The introduction is overly long and occupies too much space, while the method section is too brief and lacks sufficient implementation details, making it difficult to reproduce the work. Additionally, the topic may be too narrow in scope and might not appeal to the broader ICLR readers.
>
> We have reorganized the Introduction, added more implementation details for the Method section, and added additional details elsewhere to improve clarity. Additionally, the Appendix has more other details.
>
> Reproducibility should not be an issue, as we provide the complete code and all datasets used in our experiments.
>
> The public dataset link is available at:
> - Badminton: https://github.com/wywyWang/CoachAI-Projects/tree/main/ShuttleSet
> - T-Drive: https://www.kaggle.com/datasets/arashnic/tdriver
> - Rome: https://ieee-dataport.org/open-access/crawdad-romataxi
>
> We have already uploaded the full source code and all datasets used in our experiments as a compressed package in the Supplementary Material(OpenReview). An anonymous GitHub repository is also provided here: https://anonymous.4open.science/r/TRIDENT-D877
>
> ---
>
> ### **Question 3**: Since discrete trajectories are the main focus of the paper and show the most performance improvement, the evaluation should include more relevant datasets, such as basketball or soccer trajectories, to better validate the generality of the proposed approach.
>
> According to our definition of discrete versus continuous movement above (Response to Reviewer 719K Weakness 1/Question 1), the trajectories in sports such as soccer and basketball fall more on the continuous side. Regarding the request to include additional sports (e.g., soccer or basketball). This work focuses on dynamic, and discrete movements (Response to Reviewer 719K Question 2). Among such movements, Badminton represents one of the most extreme and challenging cases: rapid directional changes, and frequent reversals within very short time intervals. In contrast, soccer trajectories are behaviorally closer to taxi movements, generally smoother, and guided by explicit tactical objectives. As shown in Figure 6 and Table 1, the learned embeddings accurately capture path similarity and distance variations, suggesting that our model would not face additional modeling challenges on continuous sports.
>
> On the other hand, it features multi-player interactions, and a small playing area(more dynamics), which align closely with the types of data we plan to explore next. We have already included such multi-agent, variable sports trajectories in our upcoming experiments (e.g., Basketball [12] dataset link: https://github.com/sealneaward/nba-movement-data), and we hope to include the corresponding results by December 3. However, based on the reasons outlined above, we believe that using Badminton as a challenging and representative benchmark for discrete trajectories is appropriate, and we hope our ongoing extensions of the framework will be positively received. The Badminton dataset used in this study is publicly available (Response to Reviewer 719K Weakness 3).
>
> [12] Hauri et al. Multi-Modal Trajectory Prediction of NBA Players https://ieeexplore.ieee.org/document/9423150/

---

> > ### Comment · Reviewer_719K · 2025-11-21
> >
> > I appreciate the authors' response. The explanation of the properties of trajectories is helpful. However, there is still a lack of a clear mathematical definition and distinction between these two types of trajectories. Additionally, the properties seem specific to badminton trajectories. The generalization of the proposed method is limited, which might reduce its appeal to the broader ICLR audience.

---

> > > ### Author Response · Authors · 2025-11-24
> > >
> > > Regarding your comment that “the properties seem specific to badminton trajectories. The generalization of the proposed method is limited,” we sincerely appreciate your observation. We provide further clarification below.
> > >
> > > First, although the paper uses badminton as the primary example, the trajectory properties we emphasize are not exclusive to badminton. Many sports with similar structures exhibit comparable characteristics. For instance, tennis movement also shows moderate tortuosity, movement in unconstrained free space, and characteristics in discrete trajectory. Therefore, badminton trajectories should not be viewed as an isolated special case that does not generalize to other domains.
> > >
> > > Furthermore, we would like to stress that **trajectory data should not be considered as a binary categorization problem**. In practice, most trajectories lie along a spectrum of properties. Our description of two extreme cases is only meant to help readers understand differences across multiple properties—(1) spatial constraints, (2) discrete vs. continuous behaviors, and (3) tortuosity—though the properties are not limited to these.
> > >
> > > Different trajectory sources lie at different properties along these spectra. For example:
> > > - **Taxi trajectories**: (1) constrained by spatial topology, (2) continuous, (3) low tortuosity.
> > > - **Basketball or soccer trajectories**: (1) no spatial topology constraints, (2) continuous, (3) moderate tortuosity.
> > > - **Badminton or tennis trajectories**: (1) no spatial topology constraints, (2) discrete, (3) moderate–high tortuosity.
> > >
> > > This further supports that our method is **not specialized for badminton**.
> > >
> > >
> > > If what the reviewer expects is a mathematical definition, then this would correspond to formalizing trajectory distinctions via measures such as curvature complexity or other geometric metrics. In this context, we also reference [13], which analyzes sports glide trajectories using the Curvature Compression Ratio (CCR) to reflect curvature magnitude.
> > >
> > > We adopt the following formulation: $Average~CCR = \frac{1}{N} \sum_{i=1}^N\frac{L_i}{D_i}$,
> > > where $L_i$​ is the length of the $i$-th segment, $D_i$ is the distance between the $i$-th segment start and end points, and $N$ is the total number of segments.
> > > - When $Average CCR = 1$, the trajectory is perfectly straight.
> > > - When $Average CCR > 1$, the trajectory is more curved and deviates more from a straight line; a larger value indicates higher tortuosity.
> > >
> > > In Figure 9, the Average CCR for badminton trajectories and taxi trajectories (T-Drive) differ substantially—badminton has a much higher mean Average CCR and a significantly longer right tail in the distribution. This demonstrates that badminton trajectories are far more tortuous and represent a fundamentally different trajectory type compared to taxi trajectories.
> > >
> > > We have added mathematical properties and definitions in Appendix D.2 in the revised version. However, even with more formal geometric metrics, trajectory properties still do not form a strict binary categorization—this is precisely why we use two extreme cases to illustrate a broader trajectory space.
> > >
> > > Lastly, may we kindly ask: In which aspect does the reviewer believe the proposed method is limited? We would appreciate clarification so we can improve the paper accordingly.
> > >
> > > [13] Yu et al. Sliding Performance Evaluation with Machine Learning-Based Trajectory Analysis for Skeleton. https://doi.org/10.3390/data10100153.

---

> ### Comment · Reviewer_719K · 2025-11-27
>
> I appreciate the authors' response. After reading the revised paper and reference 13, the content is now much clearer. However, I would suggest including an explanation about the non-binary nature of trajectories and the properties of different common types of trajectories. This addition would (1) help readers better understand the scope of the paper and (2) provide a clearer rationale for why Badminton was chosen as a representative discrete type for the experiments. I have raised my score. It would also be beneficial if the authors could provide additional experiments on basketball trajectories.

---

> > ### Author Response · Authors · 2025-12-02
> >
> > Dear Reviewer 719K, we thank you for the constructive suggestions. In Section 2 Preliminaries, we have added explanations of the non-binary nature of trajectories and the properties of common trajectory types, and we now present more clearly our rationale for selecting Urban Mobility and Badminton as the representative of two extreme trajectory types. Additionally, we add larger scale data experiments in the revised version, with results shown in Appendix D.7. Datasets used are updated in the Supplementary Material (OpenReview).
> >
> > We understand that due to the recent OpenReview incident on anonymity, reviewers are unable to discuss further. Nonetheless, we are grateful for your patience and recognition and appreciate increasing the score to **6**.

---

### Author Response · Authors · 2025-12-03
**Message to all Reviewers and Area Chair**

### **Message to all Reviewers and Area Chair**:

Thank you to each individual reviewer for the comments, questions, and suggestions. We have responded to each reviewer’s comments in the individual responses. Here, we would like to summarize the changes that were made for clearer clarifications, motivations, methods, and contributions. We are happy and thankful for what the reviewers brought up. Motivated by each reviewer, the new PDF now additionally contains:
- Refined properties definitions and statistical analysis (Curvature Compression Ratio) on different trajectory types.
- New statistical significance tests on frameworks (Friedman test followed by the Nemenyi post-hoc test).
- New robustness tests on training with reduced and increased data/diversity.
- New experiments on cross-domain and larger datasets (Chengdu & Xian).
- New changes to text, context, and layout (Introduction, Method, etc.), with new figures and open source dataset references added.
- Refined hyperparameter tests.

---

We thank Reviewer 719K and Reviewer d5b8 for their ultimately positive responses. We have also resolved the concerns raised by Reviewer LWGz and addressed all of Reviewer M4PH suggestions. We believe this work provides reliable predictive results and a practical, novel algorithm that will benefit the ICLR community. Given the additional experiments and revisions, we hope the updates will help clarify for the new reviewer.

---

### Meta-Review · Area_Chair_8sPv · 2026-01-03

**Summary:**

The paper introduces TRIDENT, a spatio-temporal trajectory representation framework designed to handle both continuous and discrete trajectories. Initial reviews raised concerns about novelty, clarity of definitions, dataset scope, and robustness. In particular, reviewers questioned whether the framework truly unified trajectory types, noted the lack of a clear mathematical distinction between continuous and discrete movements, and highlighted limited evaluation beyond badminton data. Additional concerns included presentation quality, reproducibility, and the absence of cross-domain transfer experiments.

The authors provided a thorough rebuttal and substantial revisions. They introduced formal definitions grounded in motor control literature, added mathematical metrics such as Curvature Compression Ratio to distinguish trajectory types, and clarified the non-binary nature of trajectories. They expanded experiments to larger datasets (Chengdu, Xian), conducted cross-domain transfer tests (T-Drive to Rome), and added robustness analyses. Code and datasets were released publicly, and presentation issues were corrected. Reviewers acknowledged these improvements, with one raising their score significantly after the revisions.

Overall, while some reservations remain about broader generality and motivation for unifying diverse domains, the paper now demonstrates clear contributions in loss design, training strategy, and attention pooling, supported by extensive empirical validation. Given the responsiveness of the authors and the strengthened clarity and scope, I recommend acceptance.

**Reviewer Concerns:**

* Reviewer 719K initially questioned the novelty and the distinction between continuous vs. discrete trajectories, and raised concerns about writing quality and dataset scope. After the rebuttal, which introduced formal definitions, mathematical metrics (Curvature Compression Ratio), and clarified the non-binary nature of trajectories, the reviewer raised their score, acknowledging improved clarity and justification.

* Reviewer M4PH highlighted the lack of cross-domain transfer experiments and robustness tests. The authors responded with additional experiments (T-Drive - Rome transfer, robustness under reduced training sets) and clarified dataset collection. Presentation issues were corrected.

* Reviewer LWGz noted that the architecture followed established patterns and questioned the motivation for unifying badminton with GPS data. Concerns about dataset size and lack of comparisons with general sequence models were also raised. The rebuttal added larger datasets (Chengdu, Xian), comparisons with Transformer/LSTM baselines, and clarified the motivation for a unified representation.

* Reviewer d5b8 praised the soundness but flagged presentation issues, lack of hyperparameter details, and insufficient baseline descriptions. The rebuttal reorganized the introduction, clarified baselines, and added statistical significance tests, addressing these concerns.

**Reviewer Scores:**

Before the rebuttal, two reviewers were positive (score 6), one was at 4, and another at 2. The most critical reviewer expressed satisfaction with the rebuttal and indicated that they had increased their rating, though the updated score is not visible to me. The authors note that this rating was raised to 6. While I cannot independently confirm the exact score, the reviewer’s comments suggest a positive shift in their assessment following the rebuttal.

The other negative reviewer did not provide a follow-up, but after carefully reviewing the rebuttal, I find that most of their concerns have been adequately addressed. It is therefore reasonable to assume that their rating could also have increased to 6.

In light of these developments, it appears that all four reviewers now view the paper favorably, with likely ratings of at least 6. Accordingly, I recommend acceptance.

---

### Decision · Program_Chairs · 2026-01-26

Accept (Poster)